# Glycinergic axonal inhibition subserves acute spatial sensitivity to sudden increases in sound intensity

Tom P Franken[1,2]\*, Brian J Bondy[3], David B Haimes[3], Joshua H Goldwyn[4], Nace L Golding[3], Philip H Smith[5], Philip X Joris[1]\*

[1]Department of Neurosciences, Katholieke Universiteit Leuven, Leuven, Belgium; [2]Systems Neurobiology Laboratory, The Salk Institute for Biological Studies, La Jolla, United States; [3]Department of Neuroscience, University of Texas at Austin, Austin, United States; [4]Department of Mathematics and Statistics, Swarthmore College, Swarthmore, United States; [5]Department of Neuroscience, University of Wisconsin-Madison, Madison, United States

**Abstract** Locomotion generates adventitious sounds which enable detection and localization of predators and prey. Such sounds contain brisk changes or transients in amplitude. We investigated the hypothesis that ill-understood temporal specializations in binaural circuits subserve lateralization of such sound transients, based on different time of arrival at the ears (interaural time differences, ITDs). We find that Lateral Superior Olive (LSO) neurons show exquisite ITD-sensitivity, reflecting extreme precision and reliability of excitatory and inhibitory postsynaptic potentials, in contrast to Medial Superior Olive neurons, traditionally viewed as the ultimate ITD-detectors. In vivo, inhibition blocks LSO excitation over an extremely short window, which, in vitro, required synaptically evoked inhibition. Light and electron microscopy revealed inhibitory synapses on the axon initial segment as the structural basis of this observation. These results reveal a neural vetoing mechanism with extreme temporal and spatial precision and establish the LSO as the primary nucleus for binaural processing of sound transients.

\*For correspondence:
tfranken@salk.edu (TPF);
philip.joris@kuleuven.be (PXJ)

**Competing interests:** The authors declare that no competing interests exist.

## Introduction

A key component of the neuron doctrine is the unidirectional propagation of action potentials, formulated as the 'law of dynamic polarization' by Cajal and van Gehuchten (*Berlucchi, 1999*; *Shepherd, 1991*). As the site where action potentials are typically initiated, the axon initial segment (AIS) has a pivotal role in this process (*Bender and Trussell, 2012*; *Kole and Brette, 2018*; *Leterrier, 2018*) and is a bottleneck where inhibition can have an 'outsized' effect on a neuron's output, as proposed for chandelier and basket cells (*Blot and Barbour, 2014*; *Nathanson et al., 2019*). Disruption of such synapses is associated with severe brain disorders (*Wang et al., 2016*), but their exact functional role in the normal brain is speculative because physiological studies of these synapses have been limited to in vitro recordings. Even the basic physiological properties of axo-axonic synapses are unclear, not in the least in cortex, where it has recently even been debated whether these synapses are excitatory or inhibitory (*Woodruff et al., 2010*). Here, we report AIS inhibition by glycinergic neurons for the first time, with a specific functional role tying together several puzzling anatomical and physiological features.

Humans are exquisitely sensitive to the spatial cues of time and intensity differences between sounds at the two ears (ITDs and IIDs; *Klumpp and Eady, 1956*; *Yost and Dye, 1988*). The classic 'duplex' account posits that these two cues operate in different frequency regions: spatial localization is subserved by ITDs for low-frequency and by IIDs for high-frequency sounds (*Strutt, 1907*).

This account dovetails with the existence of two brainstem circuits seemingly dedicated to the extraction of these cues: the MSO generates sensitivity to ITDs (*Goldberg and Brown, 1969*; *Yin and Chan, 1990*) and the LSO to IIDs (reviewed by *Tollin, 2003*). These two circuits share many components: their most salient difference is that MSO neurons perform coincidence detection on the excitatory spike trains they receive from both ears, while LSO neurons perform a differencing operation comparing net excitatory input from the ipsilateral vs. net inhibitory input from the contralateral ear.

This classical duplex account of the respective role of these two binaural nuclei does not square with striking physiological and morphological features found in the circuits converging on the LSO, including some of the largest synapses in the brain (e.g. the calyx of Held). This and other observations suggest that the LSO is not simply weighing excitation vs. inhibition toward IID-sensitivity, but is specialized for temporal comparisons between the two ears. Many studies indeed observed ITD-sensitivity of LSO neurons to a range of sounds (tones, amplitude-modulated tones, noise [*Caird and Klinke, 1983*; *Irvine et al., 2001*; *Joris, 1996*; *Joris and Yin, 1995*; *Tollin and Yin, 2005*]), but ITD-sensitivity to these sounds was weak compared to the effects of IIDs and not commensurate with the striking specializations of the LSO circuit (*Joris and Yin, 1998*). The only stimuli to which strong ITD-sensitivity was occasionally observed in LSO neurons was to electrical shocks in vitro (*Sanes, 1990*; *Wu and Kelly, 1992*) and, in vivo, to brisk changes in sound characteristics, usually referred to as 'transients'. Examples of such transients are clicks, tone onsets, and fast frequency-modulated sweeps (*Caird and Klinke, 1983*; *Irvine et al., 2001*; *Joris and Yin, 1995*; *Park et al., 1996*). High-frequency transients are generated as adventitious sounds created by the locomotion of predators or prey at close range (*Clark, 2016*; *Goerlitz and Siemers, 2007*). Behavioral experiments show that lemurs rely on such sounds to forage (*Siemers et al., 2007*), and mice choose routes to minimize the generation of such sounds (*Roche et al., 1999*). This leads to the hypothesis that detection and lateralization of these sounds was a strong evolutionary pressure for this high-frequency circuit and drove its striking temporal specializations (*Joris and Trussell, 2018*). The recent discovery that LSO principal cells have fast membrane properties and respond transiently to tones (*Franken et al., 2018*) gives extra weight to the importance of timing in this circuit.

We used in vivo and in vitro whole-cell patch clamp methods to examine ITD-sensitivity in identified LSO and MSO neurons in response to transient sounds, and found exquisite tuning in LSO but not MSO neurons. LSO principal cells show a sub-millisecond window where the contralateral ear effectively vetoes the output of the ipsilateral ear, and this is dependent on the strategic positioning of inhibitory inputs on the AIS. Moreover, effects of IIDs are such that they enhance ITD-sensitivity. Thus, for sound impulses, fast temporal differentiation is implemented in LSO, and this is a more suitable neural operation for the creation of binaural sensitivity than the coincidence-type operation in MSO. Our finding that inhibition at the AIS combines with other specializations to achieve temporal differentiation that is punctate in space and time, pulls together previously puzzling anatomical and physiological features into a single coherent view that proposes a new role for LSO principal neurons.

## Results

### Sharp ITD-sensitivity to clicks in LSO but not MSO

We obtained in vivo whole-cell recordings while presenting clicks at different ITDs in 19 LSO neurons and 11 MSO neurons. Responses to tones for these cells have been reported before (*Franken et al., 2018*; *Franken et al., 2015*). We were surprised to find sharp sensitivity to ITDs of clicks in LSO but not MSO neurons. *Figure 1A* shows sensitivity to ITDs of transients in a principal LSO neuron (IID function in *Figure 1—figure supplement 1A*). Identical impulsive sounds ('clicks') were delivered to the two ears with varying ITD. The neuron reliably fires a single spike at large negative and positive ITDs, but is completely inhibited over a sub-millisecond range near 0 μs. The resulting U-shaped tuning function has extraordinarily steep slopes (−6.5 and 4.2 spikes per click/ms); a narrow and deep trough (450 μs halfwidth) with complete suppression of spiking, and low variability. A measure of tuning, ITD-SNR (the ITD-dependent variance in spike rate divided by the total variance [*Hancock et al., 2010*]) gives a value of 0.86. *Figure 1B* shows a waterfall plot of the corresponding intracellular voltage signals. It shows an orderly progression of leading EPSP and lagging IPSP at

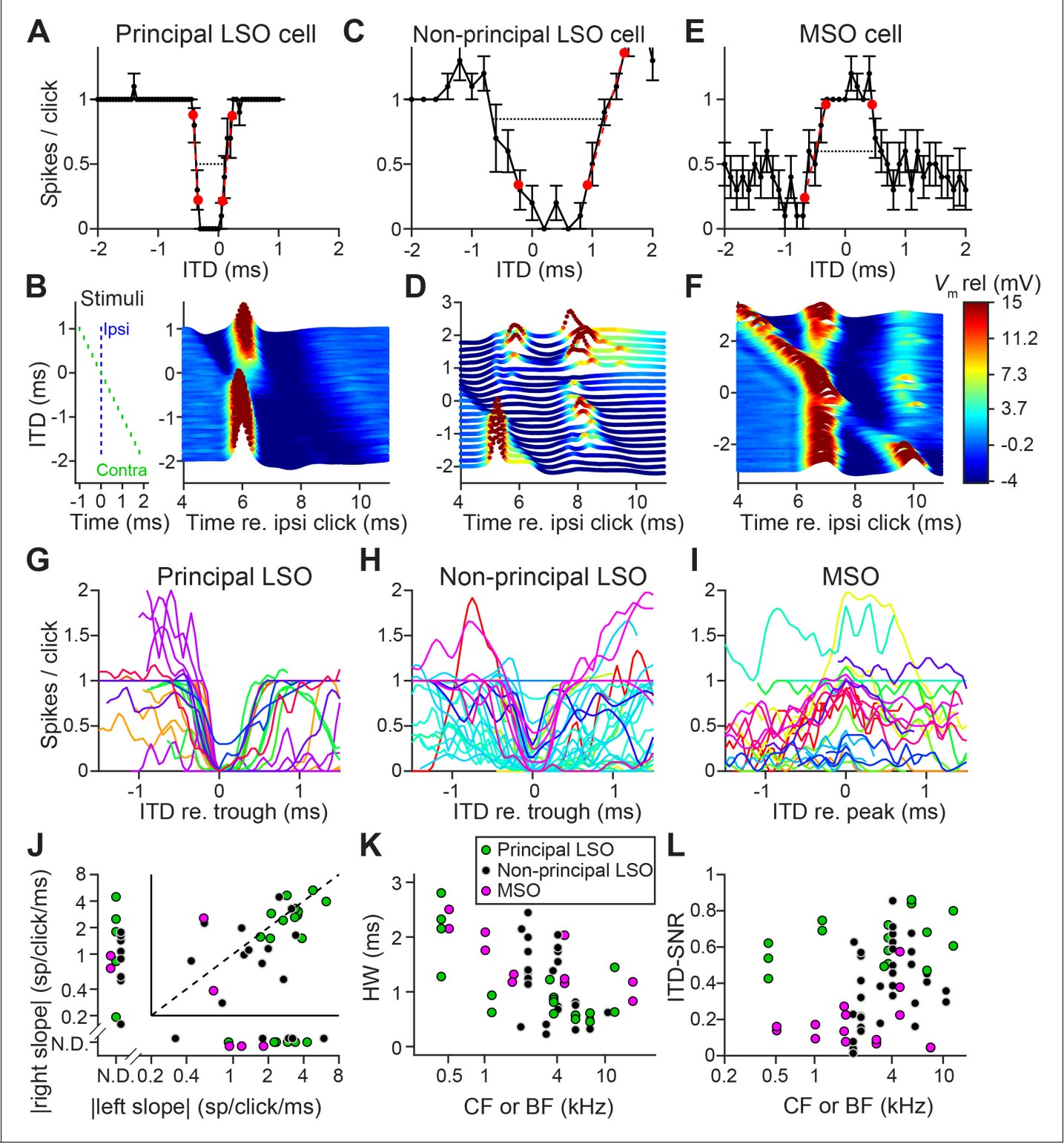

**Figure 1.** Sharp ITD-sensitivity to clicks in LSO but not MSO. (**A**) Click-ITD function (10 repetitions) of an LSO principal cell (CF = 5.7 kHz). Data are represented as mean ± SEM. Red circles indicate ITD values near the trough when the spike rate reached 20% or 80% of the maximum. Black dotted line indicates halfwidth of the central trough. By convention, negative ITD refers to the ipsilateral click leading the contralateral one, and vice versa for positive ITD. (**B**) Waterfall plots of the intracellular response of the cell in A. The membrane potential, averaged per ITD value across 10 repetitions, is color-coded. For clarity and to aid comparison with D and F, colors correspond to values clipped between the limits shown in the color scale in F. Inset on left indicates timing of the contralateral click (green) relative to the ipsilateral click (blue) for different ITDs. Leading or lagging inhibition does not

*Figure 1 continued on next page*

*Figure 1 continued*

affect the EPSP sufficiently to inhibit spiking, except over a small time window at ITDs near 0 ms. (C–F) Similar to A and B, for a non-principal LSO cell (C and D; CF = 4.1 kHz), and for an MSO neuron (E and F; CF = 4.6 kHz). In MSO, an excitatory response is present to clicks from either ear: there is some modulation of spike rate but it never decreases to 0. Red circles in E indicate ITD values where the spike rate reached 20% or 80% of the maximum. Black dotted line indicates halfwidth of the central peak. (G–I) Population of ITD functions for identified principal LSO neurons (G: 24 data sets, 8 cells), non-principal LSO neurons (H: 38 data sets, 11 cells), and MSO neurons (I: 25 data sets, 11 cells; 6 out of 11 cells were anatomically verified). To reduce clutter, tuning functions were aligned at the most negative ITD value of trough (G and H) or peak (I). Different colors indicate different cells. (J) Steepness of the slope (measured at 20% and 80% points) to the right of the central trough (for LSO cells) or peak (for MSO cells) plotted against steepness of the slope to the left of the central trough or peak. Abscissa and ordinate are scaled logarithmically. N.D.: data points for which either the left slope or the right slope is not defined because spike rate did not reach the respective threshold (e.g. the right slope of the MSO cell in E). Data sets for which both left and right slopes were not defined are not shown (LSO principal: one data set; LSO non-principal: nine data sets; MSO: 10 data sets). Only cells for which the trough was lower than 0.5 sp/click (LSO) or the peak was higher than 0.5 sp/click (MSO) were included. Principal LSO (green): 23 data sets, 8 cells; Non-principal LSO (black): 27 data sets, 8 cells; MSO (magenta): seven data sets, 4 cells. (K) Halfwidth of the central peak or trough as a function of CF or BF. Abscissa is scaled logarithmically. Only cells for which the trough was lower than 0.5 sp/click (LSO) or the peak was higher than 0.5 sp/click (MSO) were included. Principal LSO: 17 data sets, 8 cells; Non-principal LSO: 26 data sets, 9 cells; MSO: 11 data sets, 6 cells. (L) ITD-SNR (*Hancock et al., 2010*) as a function of CF or BF. Abscissa is scaled logarithmically. Principal LSO: 16 data sets, 7 cells; Non-principal LSO: 34 data sets, 9 cells; MSO: 16 data sets, 7 cells. Legend in K applies also to J and L. Numerical data represented as graphs in this figure are available in a source data file (*Figure 1—source data 1*).

The online version of this article includes the following source data and figure supplement(s) for figure 1:

**Source data 1.** Excel table with data represented in this figure.
**Figure supplement 1.** Physiological data of LSO cells in *Figure 1*.
**Figure supplement 1—source data 1.** Excel table with data represented in this figure.
**Figure supplement 2.** Population data of ITD functions of *Figure 1G–I*, without centering the left flank of the central trough (LSO) or peak (MSO) at 0 ms.
**Figure supplement 2—source data 2.** Excel table with data represented in this figure.
**Figure supplement 3.** ITD-sensitivity to clicks at different sound levels.
**Figure supplement 3—source data 3.** Excel table with data represented in this figure.
**Figure supplement 4.** ITD-sensitivity to clicks is steeper than to sustained sounds for LSO cells.
**Figure supplement 4—source data 4.** Excel table with data represented in this figure.
**Figure supplement 5.** Steep ITD-sensitivity to transients extends to rustling stimuli.
**Figure supplement 5—source data 5.** Excel table with data represented in this figure.

negative ITDs and the reverse sequence at positive ITDs, with a narrow range where the PSPs effectively oppose each other and spiking is abolished. The traces are aligned to the ipsilateral (excitatory ear) click at 0 ms (see left panel): events locked to that stimulus appear vertically stacked. As ITD changes, events locked to the contralateral ear are stacked diagonally. Clearly, the reliable response of 1 spike/click (*Figure 1A*) is in response to the ipsilateral ear. At large negative click ITDs, when the ipsilateral (excitatory) ear is leading, excitation is unopposed and reliably triggers a single spike. Likewise, at large positive ITDs, the leading contralateral (inhibitory ear) click is not able to suppress spiking to the lagging ipsilateral click, even for lags between IPSP and EPSP as small as 0.25 ms. *Figure 1C and D* show data for a non-principal LSO neuron (IID function in *Figure 1—figure supplement 1B*): here the intracellular traces are more complex than a stimulus-like stacking of PSPs, but nevertheless tuning to ITDs is present, be it with shallower slopes (−1.0 and 2.0 spikes per click/ms), wider trough (halfwidth 1350 µs), and higher variability, yielding an ITD-SNR of 0.63. *Figure 1E and F* show data for an MSO neuron. As expected, the main feature in the response is an excitatory peak near 0 ms. Even though this is one of the steepest-sloped ITD-functions of our MSO sample (2.1 spikes per click/ms for slope at ITD <0 ms), the ITD-tuning lacks the acuity observed in principal LSO cells, with an ITD-SNR of only 0.38. The intracellular data (*Figure 1F*) reveal that, surprisingly, spiking is not restricted to ITDs where the two events coincide, but also occurs at other ITDs, where the click at either ear can elicit a suprathreshold response.

Population data are shown in *Figure 1G–I* and *Figure 1—figure supplement 2*. LSO neurons were sorted into principal and non-principal neurons based on anatomical and physiological criteria (*Franken et al., 2018*). In principal cells (*Figure 1G*), the spiking output of most cells is steeply dependent on ITD at some ITD range. Non-principal LSO neuron tuning (*Figure 1H*) is much more varied but occasionally also features steep slopes. These spike data elaborate on a handful of extracellular recordings of such sensitivity in LSO (*Caird and Klinke, 1983*; *Irvine et al., 2001*; *Joris and*

*Yin, 1995*) and show that this acute temporal sensitivity is a dominant feature of principal LSO neurons, the most frequent cell type in this nucleus, which is undersampled with extracellular methods (*Franken et al., 2018*). We observed steeper slopes and narrower functions at higher sound levels (*Figure 1—figure supplement 3A,B*).

Compared to LSO, ITD-tuning was surprisingly weak in the majority of MSO neurons (*Figure 1I*). While ITD-functions of LSO neurons had steep slopes, such slopes could not be meaningfully calculated in many MSO neurons (*Figure 1J*). Halfwidths of the ITD-tuning functions, that is the ITD range over which the response is suppressed by ≥50% (for LSO), or enhanced by ≥50% (for MSO), are smaller for principal LSO cells than for MSO cells (*Figure 1K*; respective median (IQR) 0.84 ms (0.48), 8 cells, and 1.40 ms (0.74), 6 cells; Mann-Whitney $U = 40.0$, p=0.043; θ = 0.83 (95% CI [0.52, 0.95])). Halfwidth is smaller for higher log(CF) (computed across LSO and MSO cells: $r = -0.62$, p=0.001), potentially due to the non-uniform distribution of glycine receptors with higher concentrations in the high-frequency region (*Sanes et al., 1987*) and wider bandwidth at high CFs. Halfwidth does not fully capture the difference in tuning quality that can be observed when comparing *Figure 1A and G* with *Figure 1E, 1I*: tuning functions are noisier for MSO neurons than for principal LSO neurons. To capture the reliability of tuning better, we calculated ITD-SNR (Methods). ITD-SNR was substantially higher for principal LSO cells than for MSO cells, across the range of frequency tuning (*Figure 1L*; respective median [IQR] 0.62 [0.17], 7 cells, and 0.15 [0.10], 7 cells); Mann-Whitney $U = 49.0$, p=0.0006; θ = 1 (95% CI [0.73, 1]); there was no significant difference in CF for the principal LSO and MSO neurons included in this analysis (Mann-Whitney $U = 31.0$, p=0.46). We also find a significantly higher ITD-SNR for principal LSO cells than for non-principal LSO cells (non-principal LSO: median (IQR) 0.33 (0.31), 9 cells; Mann-Whitney $U = 54.0$, p=0.016; θ = 0.86 (95% CI [0.57, 0.96])). This is not explained by differences in sound level (*Figure 1—figure supplement 3C,D*).

Thus, despite the classical role of MSO neurons as 'ITD detectors', principal LSO neurons show superior ITD-tuning for transient sounds. We also tested LSO neurons with broadband noise, which has a flat amplitude spectrum like clicks but with a random phase spectrum. ITD-sensitivity to noise was generally weak (*Figure 1—figure supplement 4A*), which was also the case for responses to dynamic interaural phase differences in modulated or unmodulated pure tones (*Figure 1—figure supplement 4B*). However, the presence of brisk transients in sustained sounds, for example at tone onset (*Figure 1—figure supplement 4C*) could lead to sharp ITD-sensitivity, as was also the case for a succession of transients, simulating rustling sounds (*Ewert et al., 2012*; *Figure 1—figure supplement 5*). Thus, a broad stimulus spectrum does not suffice, and a brief duration followed by silence is not required for the generation of sharp ITD-sensitivity: the necessary and sufficient condition is to have fast and large changes in stimulus amplitude. Inspection of the membrane potential traces revealed why transients are more effective than sustained sounds: EPSPs evoked by transients are steeper than those evoked by ongoing sounds, with lower action potential voltage thresholds, and IPSPs are often steeper as well (*Figure 1—figure supplement 4D,E*).

## Effective inhibition of LSO neurons is limited to a short initial part of the IPSP

Prior to our recordings, published LSO in vivo intracellular recordings were limited to a few traces (*Finlayson and Caspary, 1989*). To gain insight into the sharp ITD-tuning in LSO and its lack in MSO, we compared intracellular synaptic responses to monaural and binaural clicks from these neurons (*Figures 2* and *3*). As illustrated for two LSO neurons (*Figure 2A and B*), they receive a well-timed EPSP in response to monaural ipsilateral clicks, which reliably trigger spikes, and a well-timed IPSP in response to monaural contralateral clicks. There have been many indirect estimates of the effective latency of excitation and inhibition in LSO neurons using in vivo extracellular recording, suggesting a close match between the onset of EPSPs and IPSPs, despite the longer pathway and extra synapse for contralateral inhibition. For example, tuning curves centered at negative ITDs (*Figures 1A* and *2D*) suggest that contralateral inhibition effectively has a shorter latency (by a few hundred microseconds) than ipsilateral excitation, while the opposite is the case when centered at positive ITDs (*Figures 1C* and *8L*). Our intracellular records allow direct measurement and show that indeed the latencies are closely matched, for both principal and non-principal neurons, with the IPSP sometimes arriving first (*Figure 2C*). In eight principal neurons, we observed a small positive deflection preceding the IPSP by ~0.5 ms (arrowhead and insert *Figure 2A*, see also *Figure 2F* and

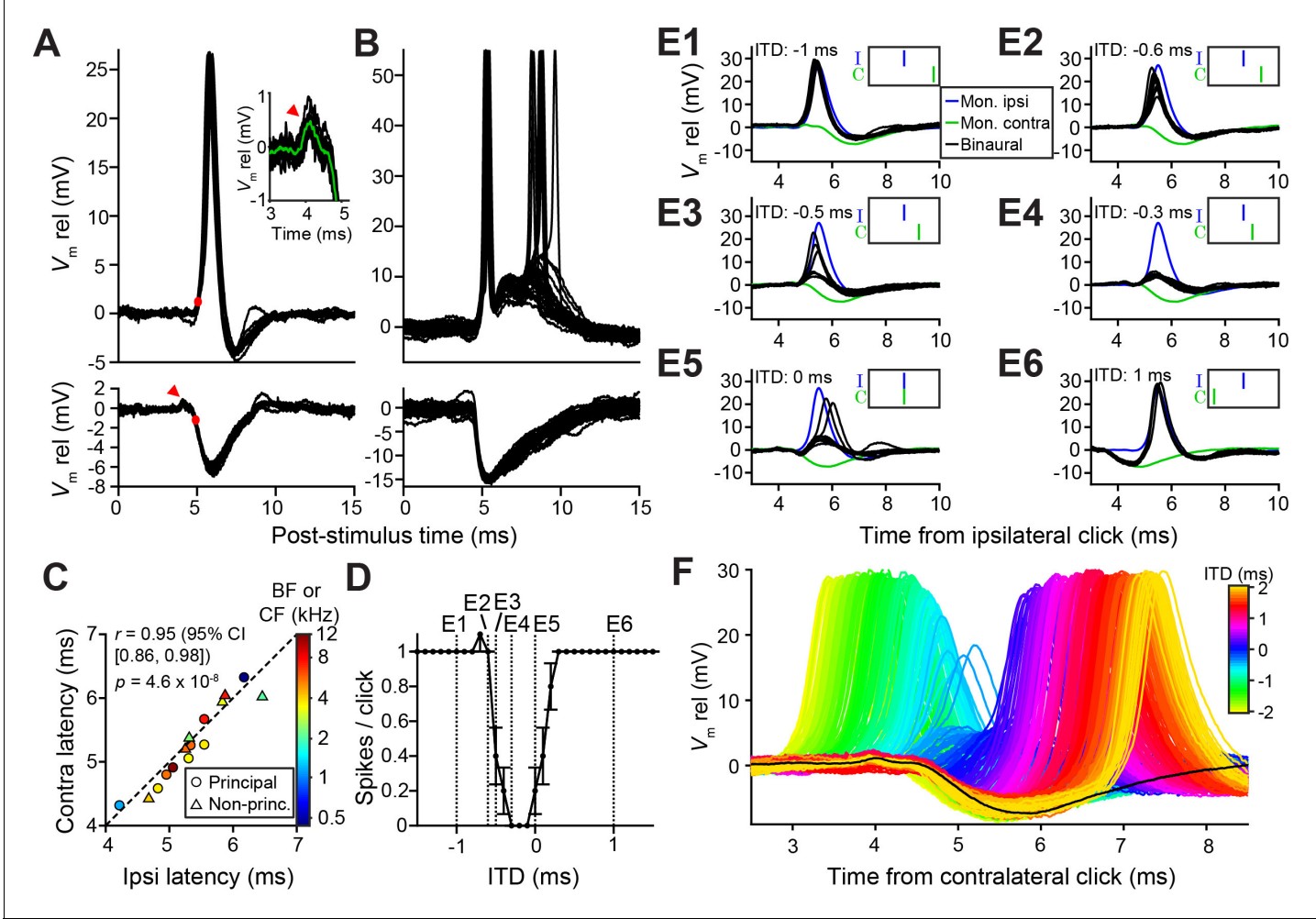

**Figure 2.** Precise timing and interaction of IPSPs and EPSPs in LSO neurons. (**A–B**) Example responses to ipsilateral clicks (top panels) and contralateral clicks (bottom panels) for a principal LSO cell (A; CF = 12 kHz; 50 dB SPL; 10 repetitions) and a non-principal LSO cell (B; CF = 4.1 kHz; 20 dB SPL; 30 repetitions). Red arrowhead indicates prepotential, shown with a blowup in inset. (**C**) Ipsilateral versus contralateral latency of postsynaptic potentials evoked by monaural clicks for nine principal cells (circles) and six non-principal LSO cells (triangles). Color indicates CF or BF for each cell. Latency was defined as the time relative to click onset when the membrane potential crossed a voltage difference relative to rest with an absolute value equal to 20% of the IPSP amplitude. For the ipsilateral response, this voltage difference was a depolarization (red dot in top panel in A), for the contralateral response this voltage difference was a hyperpolarization (red dot in bottom panel in A). For this analysis, we used the response to identical sound levels for ipsi and contra, at the lowest level generating a maximal monaural ipsilateral response. (**D**) Click-ITD function for the same neuron as in A. Data are represented as mean ± SEM. Sound level 60 dB SPL at both ears. Dotted vertical lines correspond to the ITD values of the panels in E. (**E1–E6**) Average responses to monaural ipsilateral (blue) and monaural contralateral (green) clicks are compared to binaural responses (black) for the ITD values indicated by dotted vertical lines in D. (**F**) Data from the same principal LSO cell as in A,D,E. Colored lines: responses to click pairs of different ITDs, referenced in time to the contralateral (inhibitory) click. Black line: averaged response to contralateral clicks at the same sound level as in D and E (corresponding to green line in E). Numerical data represented as graphs in this figure are available in a source data file (*Figure 2—source data 1*). The online version of this article includes the following source data and figure supplement(s) for figure 2:

**Source data 1.** Excel table with data represented in this figure.

**Figure supplement 1.** Precise interaction of IPSPs and EPSPs for another principal LSO neuron.

**Figure supplement 1—source data 1.** Excel table with data represented in this figure.

*Figure 2—figure supplement 1C*), suggesting that consistent, precise response timing is already present at the presynaptic level.

Strikingly, the IPSP duration extends to almost 5 ms in principal cells (*Figure 2A*), close to an order of magnitude larger than the halfwidth of the tuning function to ITDs (*Figures 1A* and *2D*). This is consistent with in vitro data (*Sanes, 1990*; *Wu and Kelly, 1992*), where the effective

window of inhibition was also reported to be much smaller than the IPSP duration. The availability of the monaural responses allows us to examine this window. *Figure 2E* shows comparisons of binaural and monaural click responses, for a principal LSO neuron (ITD-function in *Figure 2D*). In each panel, the intracellular response is shown at one ITD (black traces), with the responses to monaural ipsi- (blue) and contralateral clicks (green) superimposed, incorporating the stimulus ITD. At large negative delays (*Figure 2E1 and 2E2*), the leading EPSP reliably triggers spiking, unhindered by the ensuing IPSP. More surprisingly, when the IPSP leads and significantly overlaps with the EPSP (*Figure 2E6*), it also fails to inhibit spiking. Only when the early steep slope of the IPSP coincides with the early steep slope of the EPSP are spikes completely blocked (*Figure 2E4*). Comparison of binaural responses for a fuller range of ITDs with the monaural IPSP, is shown in *Figure 2F*. Responses from large negative to large positive ITDs reveal the exceedingly narrow range of ITDs over which spikes are suppressed, near the onset of the IPSP. *Figure 2—figure supplement 1* shows another example.

## Weak tuning in MSO neurons results from a breakdown of coincidence detection for transients

Ideal coincidence detectors are akin to multipliers: they only generate an output spike when receiving a spike from each input. MSO neurons approach archetypal coincidence detectors (*Joris and van der Heijden, 2019*) and respond poorly to monaural stimulation and to temporally misaligned inputs (*Goldberg and Brown, 1969*; *Yin and Chan, 1990*). This largely failed in response to clicks, because of a surprising efficacy of monaural stimuli. *Figure 3A and 3B* show responses to monaural ipsilateral (top panels) or contralateral (bottom panels) clicks for two MSO neurons. Depolarizing events dominate and often generate spikes. Examples of spike rates as a function of click intensity (*Figure 3C and 3D*), illustrate that monaural spike rates to clicks were substantial and could even equal spike rates to binaural clicks (here delivered at ITD = 0 ms, generating a spike rate >90% of the peak of the click ITD function). We calculated the summation ratio (*Goldberg and Brown, 1969*), that is the ratio of the spike rate to binaural stimulation to the sum of monaural responses, where values > 1 indicate facilitation, as expected for a coincidence detector. MSO summation ratios in response to clicks (*Figure 3F*, magenta) were all <1.3 and had a median of 0.97, indicating that the binaural response rates are similar to the sum of monaural response rates. In contrast, MSO summation ratios to tones were substantially higher than those to clicks (*Figure 3F*, black; respective median (IQR) 7.56 (11.41), 23 cells and 0.97 (0.38), 5 cells; Mann-Whitney $U$ = 4.0, p=0.001; θ = 0.03 (95% CI [0.004, 0.29])). This convincingly shows that the binaural advantage that MSO neurons display for tones is largely non-existent for clicks.

LSO neurons can be regarded as 'anti-coincidence' detectors, where the binaural rate can drop to 0 spikes and the response to monaural ipsilateral stimulation saturates near one spike/click (*Figure 3E*; double events were sometimes observed, *Figure 3—figure supplement 1*). Contralateral stimulation does not generate spiking except sometimes at high stimulus intensities, presumably due to acoustic crosstalk. To calculate the LSO neuron summation ratio, we invert the ratio (sum of monaural response rates/binaural response rate): a lack of binaural effect again results in a summation ratio of one, and binaural interaction results in larger ratios. LSO responses all resulted in summation ratios well above two (*Figure 3F*, green), substantially higher than for MSO responses to clicks (respectively 7 cells and 5 cells, Mann-Whitney $U$ = 0, p=0.003; θ = 0 (95% CI [0, 0.32])). Binaural summation to clicks for LSO cells thus clearly surpasses that of MSO cells.

## In vitro recordings reveal powerful inhibition for synaptically evoked but not for simulated IPSPs

To better understand the narrow, sub-millisecond (*Figures 1A, 1K* and *2D*) window of inhibition in LSO neurons, we performed in vitro experiments in P19-22 gerbils. Shocks to LSO afferents evoke well-timed, transient inhibition and/or excitation and are a particularly apt analogue of acoustic clicks. Thus, comparison of in vitro and in vivo data is unusually straightforward because the afferent signals evoked by shock and clicks are similar to a degree that is only rarely achieved in this type of comparison. Indeed, early experiments shocking inputs on both sides (*Sanes, 1990*; *Wu and Kelly, 1992*), and recent experiments using optogenetic stimulation (*Gjoni et al., 2018b*), showed clear ITD-sensitivity very similar to in vivo responses. *Figure 4A* shows recordings from an LSO neuron

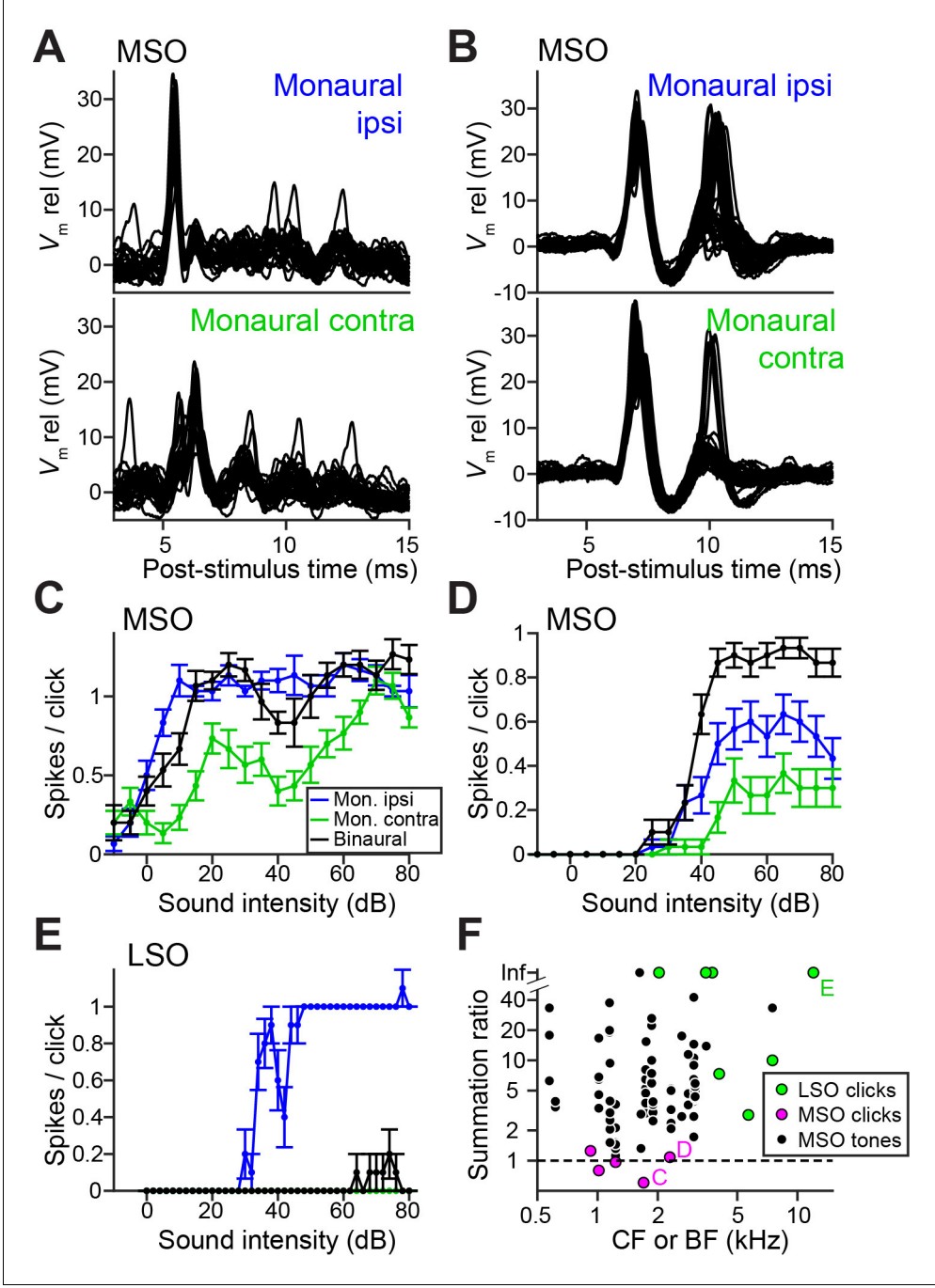

**Figure 3.** Weak ITD-tuning in MSO neurons results from a breakdown of coincidence detection for transients. (**A**) Top panel: Example responses to ipsilateral clicks (top panel) and contralateral clicks (bottom panel) for an MSO cell (CF = 1.8 kHz). Sound level 70 dB SPL. 30 repetitions shown. (**B**) Similar to A, for another MSO cell (CF = 4.6 kHz). Sound level 70 dB SPL. 30 repetitions shown. (**C**) Rate-level functions (30 repetitions) for monaural clicks and for binaural clicks at 0 ITD for the same MSO cell as in A. Data are represented as mean ± SEM. (**D**) Similar to C, for another MSO cell (CF = 2.3 kHz). Thirty repetitions per SPL. (**E**) Similar to C and D, for a principal LSO cell (CF = 12 kHz). Thirty repetitions per SPL. (**F**) Summation ratio for LSO responses to clicks (seven data sets, seven cells), MSO responses to clicks (five data sets, five cells) and MSO responses to sustained tones (77 data sets, 22 cells). For a summation ratio of one (dashed horizontal line), the binaural response equals the sum of the monaural responses. Letters C,D,E indicate data points of the cells in the corresponding panels. For MSO responses to tones, one data point with a summation ratio of ~200 is not shown. Numerical data represented as graphs in this figure are available in a source data file (*Figure 3—source data 1*).

*Figure 3 continued on next page*

*Figure 3 continued*
The online version of this article includes the following source data and figure supplement(s) for figure 3:
**Source data 1.** Excel table with data represented in this figure.
**Figure supplement 1.** Monaural stimulation often leads to double events both in LSO and MSO.
**Figure supplement 1—source data 1.** Excel table with data represented in this figure.

where ipsilateral excitation was triggered synaptically by electrical shocks, inhibition was simulated by injecting simulated conductances, and their relative timing was varied to mimic ITDs. Surprisingly, this protocol did not result in a profound inhibitory trough in the tuning function in vitro (***Figure 4C***). Results for seven other neurons are shown in ***Figure 4—figure supplement 1D*** (solid lines): although

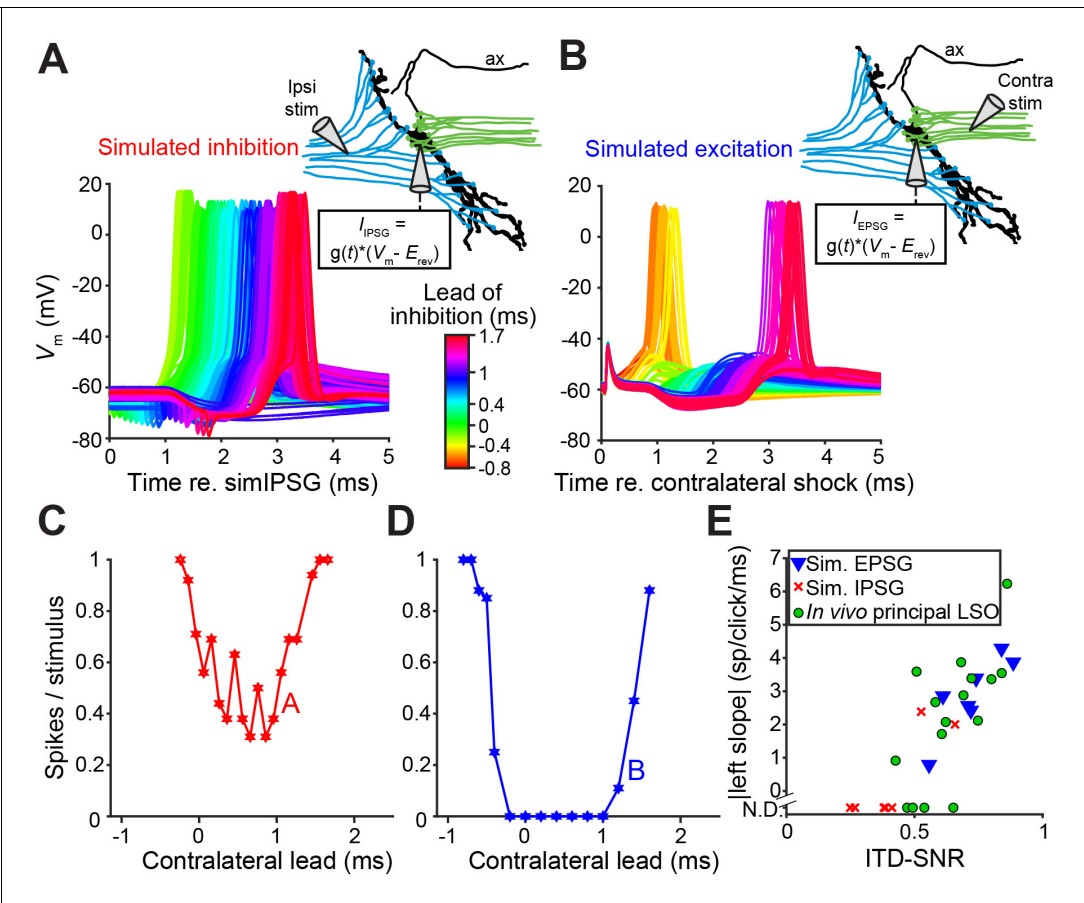

**Figure 4.** In vitro recordings reveal powerful inhibition for synaptically evoked but not for simulated IPSPs. (**A**) Voltage responses from a principal LSO neuron recorded in a brain slice, for which the ipsilateral inputs were activated by electric shocks and the contralateral input was simulated by somatically injecting inhibitory conductances (IPSG) via dynamic clamp. Inset indicates experimental setup. Delay between ipsilateral shock and IPSG was varied and is referenced to the timing of the simulated IPSG, and all recorded membrane potential traces are shown, color coded for the delay. ax: axon. (**B**) Similar to A, but with contralateral inputs activated by electric shocks and ipsilateral excitatory conductances simulated via somatic dynamic clamp (different neuron than A). (**C**) Rate delay function corresponding to the experiment in A. (**D**) Rate delay function corresponding to the experiment in B. (**E**) Steepness of the slope to the left of the trough for the population of delay functions (***Figure 4—figure supplement 1D and E***, solid lines), plotted against the ITD-SNR (as in ***Figure 1L***). Data from principal LSO cells recorded in vivo are shown for comparison (16 data sets from 7 cells). N.D.: not defined (slope was not defined when 20% of maximal spike rate was not reached after smoothing [see Materials and methods]). Numerical data represented as graphs in this figure are available in a source data file (***Figure 4—source data 1***).
The online version of this article includes the following source data and figure supplement(s) for figure 4:

**Source data 1.** Excel table with data represented in this figure.
**Figure supplement 1.** In vitro recordings reveal powerful inhibition for synaptically evoked IPSPs but not for IPSPs simulated by current injection.
**Figure supplement 1—source data 1.** Excel table with data represented in this figure.

U- or V-shaped tuning functions were obtained, full inhibition (spike rate of 0 spikes/s) was not reached in most cases. ITD tuning expressed as ITD-SNR is less pronounced for these in vitro functions than for in vivo functions (respective median (IQR) 0.40 (0.18), eight neurons and 0.62 (0.17), seven neurons; Mann-Whitney $U$ = 51.0, p=0.006; θ = 0.91 (95% CI [0.62, 0.98])). Thus, inhibition simulated by somatic injection of conductances cannot reproduce the profound inhibition observed with comparable levels of hyperpolarization in vivo. In dynamic clamp experiments, the conductances must be delivered to a single somatic location, in contrast to the more distributed spatial distribution of real inhibitory synapses that could be distributed along the axon hillock, soma and proximal dendrites. To test whether the efficacy of inhibition is sensitive to the spatial location of inhibition, we reversed the stimulus protocol: we generated natural, spatially distributed synaptic inhibition by delivering shocks to contralateral inputs, while excitation was simulated by conductance clamp at the soma. Under these conditions, despite the similar range of hyperpolarization apparent at the soma, profound inhibition of spiking was reached in the neuron in *Figure 4B* and in all neurons. The tuning functions were very similar to the ITD functions observed in vivo (*Figure 4D*; *Figure 4—figure supplement 1E*, solid lines); compare with *Figures 1A, 1G* and *2D*, in terms of slope as well as ITD-SNR (*Figure 4E*, blue). The difference in ITD-SNR between simulated inhibition and simulated excitation was statistically significant (respective median (IQR) 0.40 (0.18), eight neurons and 0.72 (0.18), seven neurons; Mann-Whitney $U$ = 54.0, p=0.001; θ = 0.96 (95% CI [0.69, 1])). The same findings were obtained when electrical currents instead of conductances were injected to simulate synaptic input (*Figure 4—figure supplement 1* [dashed lines]). Together, these results suggest that at least some inhibitory synapses are located electrically closer to the spike initiation region in the axon, prompting a detailed examination of the spatial pattern of inhibitory synapses at the soma, axon hillock and the AIS.

## LSO, but not MSO, neurons have inhibitory innervation of the axon initial segment

The connectivity of LSO neurons has been extensively studied (*Cant, 1991*; *Glendenning et al., 1985*; *Yin et al., 2019*). Although there are remaining questions particularly regarding the identity of input from the cochlear nucleus (*Doucet and Ryugo, 2003*; *Gómez-Álvarez and Saldaña, 2016*), the inhibitory input provided by the homolateral medial nucleus of the trapezoid body (MNTB) is well-characterized (*Banks and Smith, 1992*; *Gjoni et al., 2018a*; *Kapfer et al., 2002*; *Smith et al., 1998*). The inhibitory projection targets LSO somata (*Gjoni et al., 2018a*; *Smith et al., 1998*), so it is surprising that somatic injection of IPSGs or IPSCs does not fully mimic synaptic stimulation. That the actual inhibitory synaptic input is more powerful than somatic current injection suggests a specialization distal from the soma, possibly the AIS. To visualize the spatial pattern of glycinergic terminals on LSO neurons, we immunostained the LSO for gephyrin and ankyrin G, markers for postsynaptic glycine receptors and the scaffolding of the AIS/nodes of Ranvier, respectively (*Figure 5*). We additionally stained for DAPI or for synaptophysin-1 (SYN1) to visualize somata or synaptic boutons, respectively. We analyzed samples from the mid-frequency region of the LSO (*Figure 5A and B*), where electrophysiological recordings were typically made, selecting neurons with a complete, relatively planar AIS that could be unambiguously connected to an axon hillock and soma (*Figure 5C–H*). A high density of gephyrin-positive puncta covered the soma and proximal dendrites. In several cases, gephyrin-positive puncta could also be seen extending onto the axon hillock (*Figure 5*, filled yellow arrowheads), and/or along the AIS itself (*Figure 5*, open yellow arrowheads). Clear overlap of gephyrin-positive puncta with a putative synaptic terminal labeled by synaptophysin-1 on an uninterrupted AIS was sometimes seen (*Figure 5F*, open white arrowhead).

To obtain conclusive proof of innervation of the AIS, we performed electron microscopy (EM) on three principal LSO neurons labeled with biocytin. *Figure 6A* shows a camera lucida drawing of a principal LSO neuron, with indication of parts of the axon that were examined with EM. A section at a distance of several tens of μm from the soma, shows the myelinated axon (*Figure 6B*). A section through the AIS shows indeed three synaptic profiles (*Figure 6C*, enlarged in *Figure 6D1-3*). The same was true for the two additional principal LSO neurons (*Figure 6—figure supplement 1A*). In contrast, principal MSO neurons (n = 2) did not show such innervation (*Figure 6—figure supplement 1B*).

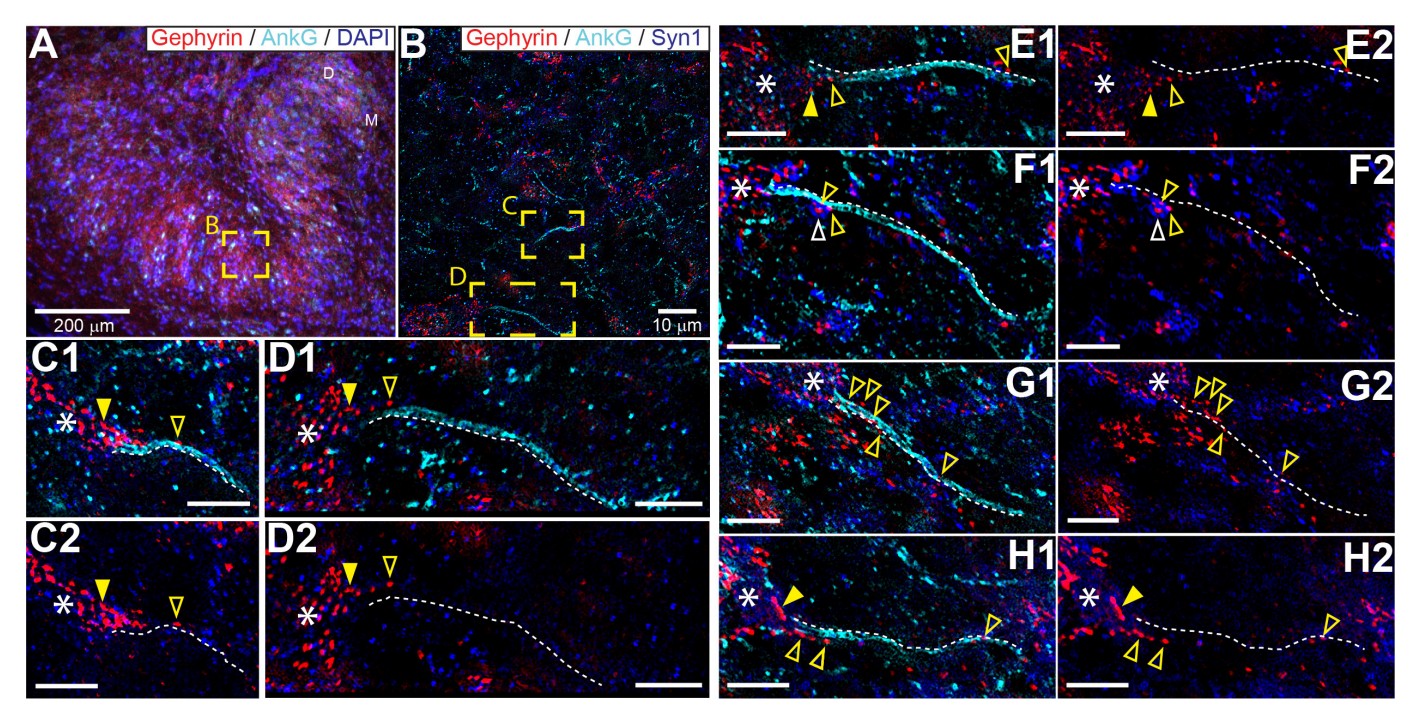

**Figure 5.** Glycinergic innervation of the axon initial segment of LSO neurons. (**A**) An example of an LSO in a coronal tissue section from the Mongolian gerbil outlining subregions targeted for SR-SIM microscopy labeled with gephyrin (red), ankyrinG (cyan), and DAPI (blue). All imaging was targeted to the middle bend of the LSO. (**B**) An example SR-SIM multichannel image labeled for gephyrin (red), ankyrinG (cyan), and synaptophysin1 (blue). Yellow boxes indicate axon initial segments (AIS) shown in C (mirrored from B) and D. (**C–H**) Images showing example LSO AISs with (1) or without (2) labels for ankyrinG channel (cyan). White dotted lines lay adjacent to labeled AIS for visual guidance, but are not quantitatively drawn. Putative inhibitory terminals can be seen closely associated with the AIS (open yellow arrowheads) and axon hillock (filled yellow arrowheads). Some large putative gephyrin positive terminals show colocalization with synaptophysin1 labeling (open white arrowheads). White asterisks indicate the soma/dendrite from which the AIS emerges. All scale bars are 5 μm, unless noted otherwise.

## Computational model shows that inhibitory synapses have a larger effect when added to the axon initial segment instead of to the soma

To test the hypothesis that adding inhibitory synapses to the AIS results in more powerful inhibition compared to only somatic inhibition, we constructed an LSO neuron model. In brief, we adapted the approach of *Goldwyn et al., 2019* to describe soma and axon regions of an LSO neuron. Soma-axon coupling was defined by coupling constants (voltage attenuation factors) and additional parameters were informed by previous models of LSO neurons (*Ashida et al., 2017*; *Gjoni et al., 2018a*; *Wang and Colburn, 2012*), typical response characteristics such as membrane time constant and input resistance (*Sanes, 1990*), and features of our in vivo recordings (such as amplitude and variability of inhibitory post-synaptic potentials). See Materials and methods and *Figure 7—figure supplement 1* for further details.

We created ITD tuning curves by computing spike probability as we varied the time lag between excitatory and inhibitory inputs (positive ITD if inhibition leads, see voltage traces in *Figure 7B*). We selected parameter values so that the model exhibited modest ITD tuning when all eight inhibitory synapses contacted the soma (minimum spike probability of approximately 0.5 for the dashed line in *Figure 7A*). This outcome matched the relatively shallow ITD tuning curves measured in vitro when IPSPs were delivered through dynamic clamp or current injection (*Figure 4*, *Figure 4—figure supplement 1*). The ITD tuning curve became substantially deeper when we relocated two of the inhibitory synapses and placed them on the AIS (solid line in *Figure 7A*). This reflects the combined effect of soma and AIS-targeting inhibition: the AIS inhibition alone was not sufficient (compare solid and dotted line in *Figure 7A*). These computational results thus provide support for our hypothesis that

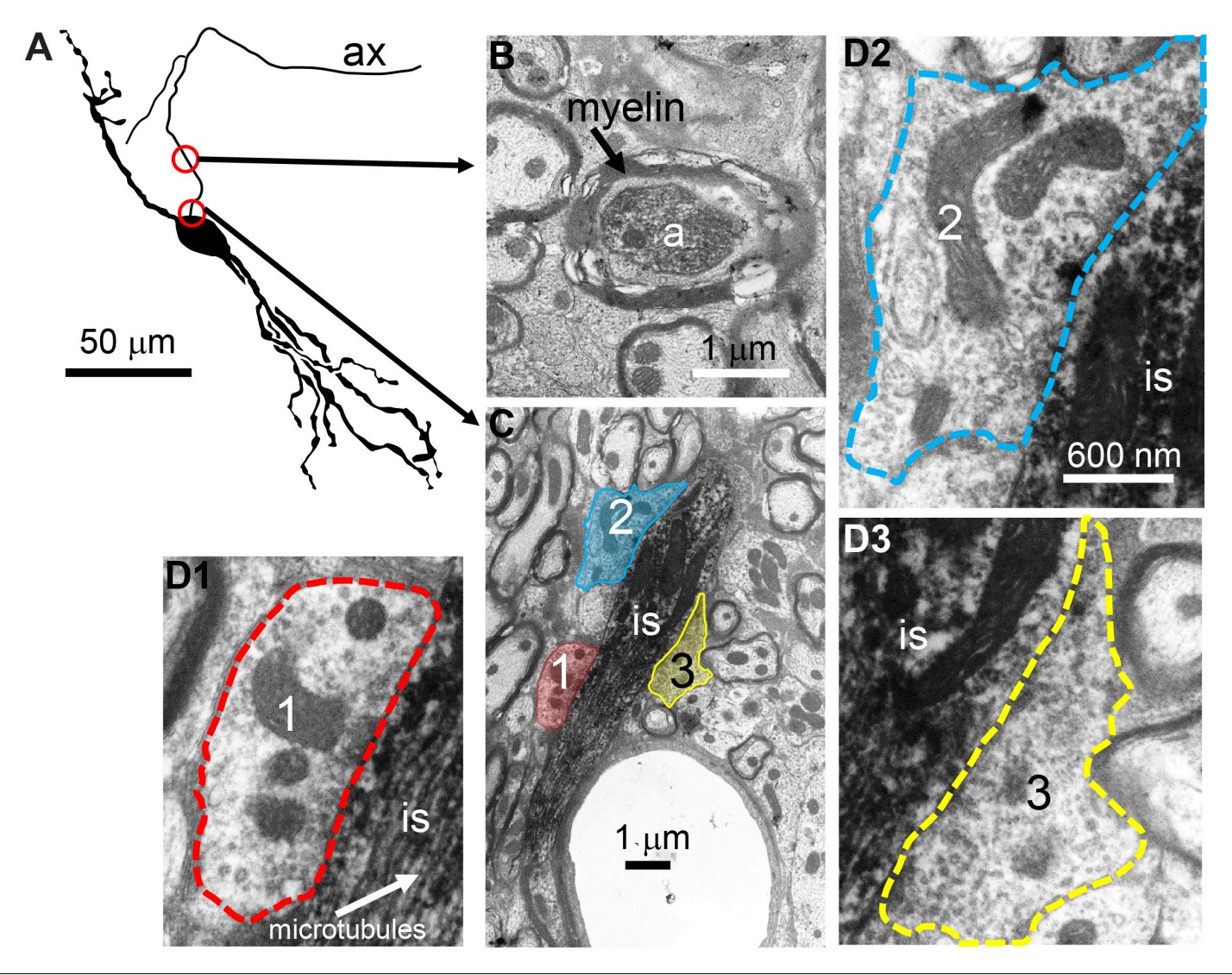

**Figure 6.** Electron microscopy reveals synaptic terminals on an LSO principal cell's axon initial segment. (**A**) Camera lucida drawing of an LSO principal cell that was intracellularly recorded from and labeled, in vivo. This cell corresponds to cell two in *Franken et al., 2018*, their Figure 2A. Arrows point to electron micrographs that show portions of the axon (ax) enclosed by the red circles. (**B**) Electron micrograph showing portion of the axon in the top circle in A. The axon is myelinated here. (**C**) Electron micrograph showing portion of the axon in the bottom circle in A. This is at the level of the axon initial segment (is). Enclosed colored areas 1–3 represent axon terminals synapsing on the axon initial segment. (**D1**–**D3**) Electron micrographs showing larger versions of axon terminals 1–3 in C. Scale bar in D2 applies to all three enlarged micrographs.

The online version of this article includes the following figure supplement(s) for figure 6:

**Figure supplement 1.** Electron microscopy reveals synaptic terminals on the axon initial segment of principal LSO cells but not of principal MSO cells.

a combination of inhibition at the soma and the AIS is more powerful than inhibition restricted to the soma, and can result in steep ITD functions in LSO neurons.

To understand why inhibitory synapses on the AIS have a larger effect than the same number of synapses on the soma, we can compare how these two inhibition sources impact voltage in the AIS region. Consider a simplified scenario of steady-state responses to constant current inputs (the same argument can be modified for dynamic inputs using frequency-dependent impedance functions). In this case, soma-targeting inputs affect soma voltage in proportion to soma input resistance $R_1$ and this voltage spreads to the AIS in proportion to $\kappa_{1\rightarrow2}$ (see discussion of coupling constants in Materials and methods, and also *Goldwyn et al., 2019*). The impact of soma-targeting inhibitory input on AIS voltage is, therefore, proportional to $R_1\kappa_{1\rightarrow2}$ (equivalent to the transfer resistance from

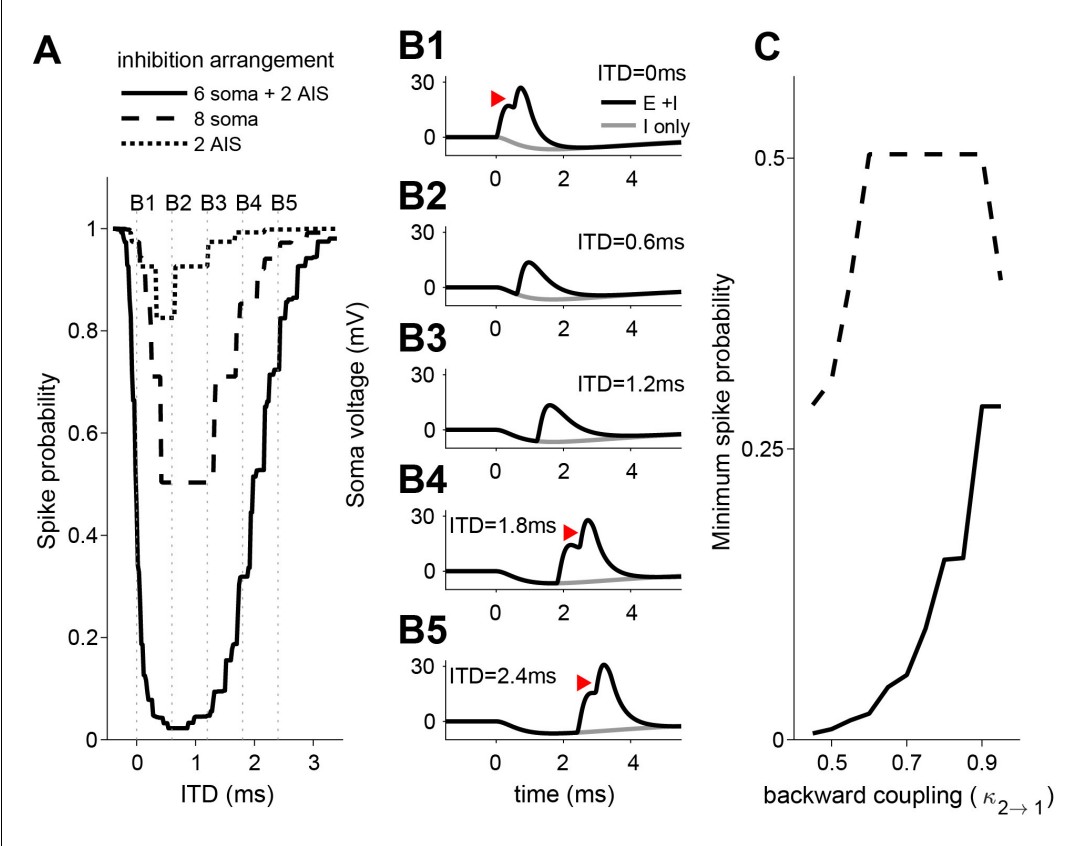

**Figure 7.** ITD-tuning in a two-compartment LSO neuron model. (**A**) ITD-tuning is substantially deeper when inhibitory inputs target AIS as compared to the same total number of inputs targeting the soma only. (**B**) Soma voltage showing detailed timing of responses to excitatory and inhibitory inputs. All synaptic inputs are activated in these simulations. Inhibition arrangement is six soma and two AIS inputs. Coupling configuration in **A** and **B** is $\kappa_{1\rightarrow2}$ = 0.95 and $\kappa_{2\rightarrow1}$ = 0.6. Red arrowheads mark action potentials (backpropagated from initiation site in the AIS compartment). Voltage is expressed relative to rest. (**C**) More pronounced suppression of spike rates when inhibitory synapses are added to the AIS rather than to the soma occurs for a range of backward coupling values. Inhibition arrangement for the two functions is indicated in the legend in **A**. Forward coupling in all simulations is $\kappa_{1\rightarrow2}$ = 0.95. Numerical data represented as graphs in this figure are available in a source data file (**Figure 7—source data 1**).

The online version of this article includes the following source data and figure supplement(s) for figure 7:

**Source data 1.** Excel table with data represented in this figure.
**Figure supplement 1.** Two-compartment LSO neuron model.
**Figure supplement 2.** ITD tuning in two-compartment LSO neuron model with synaptic kinetics adapted from *Beiderbeck et al., 2018*.
**Figure supplement 2—source data 1.** Excel table with data represented in this figure.

soma to AIS [*Koch et al., 1982*]). In contrast, inhibitory inputs on the AIS change AIS voltage in proportion to AIS input resistance $R_2$. Thus, we expect AIS-targeting inhibition to be more powerful than soma-targeting inhibition if $R_2 > R_1\kappa_{1\rightarrow2}$. This condition is always satisfied because $R_2\kappa_{2\rightarrow1} = R_1\kappa_{1\rightarrow2}$ since both terms are expressions for the transfer resistance (discussed in Materials and methods) and $\kappa_{2\rightarrow1} < 1$ under the plausible constraint that voltage must attenuate somewhat from AIS to soma. For the simulations in *Figure 7A,B*, for instance, we used $\kappa_{1\rightarrow2}$ = 0.95, $\kappa_{1\rightarrow2}$ = 0.6, $R_1$ = 40 MΩ, and $R_2$ = 64 MΩ. We found qualitatively similar results using a range of backward coupling constants (*Figure 7C*) and found that ITD-tuning depth is relatively insensitive to changes in forward coupling (results not shown).

To test whether our findings are robust for waveform kinetices different than those used in the dynamic clamp experiments, we performed the same analyses using conductance waveforms adapted from a recent study of mature LSO neurons (*Beiderbeck et al., 2018*; Materials and methods). The findings are qualitatively similar, with narrower tuning functions due to faster IPSGs (*Figure 7—figure supplement 2*).

## LSO neurons show graded latency-intensity changes which disambiguate spatial tuning

It has been hypothesized that temporal specializations in the LSO-circuit evolved to generate tuning to ITDs of transient sounds congruent with IID-tuning (*Joris and Trussell, 2018*). Classical tuning to IIDs is sigmodial (*Figure 1—figure supplement 1A* and *Figure 1—figure supplement 1B*), with higher spike output for IIDs < 0 and complete inhibition of spiking for IIDs > 0, so that LSO neurons are excited by sounds in the ipsilateral hemifield (*Tollin and Yin, 2002*; *Figure 8A*, cartoons below the abscissa illustrate the accompanying PSP changes). Congruence of ITD- and IID-tuning would be obtained if the 'left' slope of ITD-functions is centered over the ITD-range relevant to the animal (*Figure 8B*, function 3): an increase in firing rate would then consistently signal a sound source more toward the ipsilateral side, for both cues. Our sample (*Figure 1 Figure 1—figure supplement 2*), as well as published ITD-functions (*Beiderbeck et al., 2018*; *Irvine et al., 2001*; *Joris and Yin, 1995*; *Park et al., 1996*), do not support such congruency as a dominant feature: indeed for at least a sizable fraction of neurons, it is the 'right' slope that is closest to 0 ms (*Figure 8B*, function 1). For cases where the ITD-function is centered near 0 ms (*Figure 8B*, function 2, example in *Figure 8L* [cyan]), there is an additional issue of ambiguity: a rise in spike rate could signal both a leftward or rightward change in horizontal position of the sound source. A similar problem occurs at the population level if some neurons have the 'left' slope near 0 and others the 'right' slope. However, a natural and elegant solution to these issues is directly embedded in the properties of the LSO circuit.

*Figure 8D and E* show how PSPs change with sound intensity for a principal and a non-principal cell. In the principal neuron, the changes in both EPSP and IPSP are extremely reproducible and finely graded in amplitude and latency with increasing SPL, also for individual trials (*Figure 8—figure supplement 1A*). In the non-principal neuron, the changes are complex, with multiple events following each click. The latency changes are sizeable compared to the relevant ITD range for the animal: they show a steady decrease which is approximately linear over the 30 dB range tested, with a slope amounting to ~10–20 μs/dB (*Figure 8F*: 13 μs/dB ipsi and 20 μs/dB contra; *Figure 8G*: 10 μs/dB ipsi and contra).

In real-world environments, IIDs and ITDs co-occur and are correlated (*Gaik, 1993*). For transient stimuli, the two cues merge into a single EPSP – IPSP pair with a given amplitude and time difference. *Figure 8H–J* use monaural responses to characterize such pairs for variations of single or combined cues. For changes in ITD only (IID fixed at 0 dB), three pairings are shown (*Figure 8I*). The spike rates obtained for these conditions are indicated in *Figure 8L* (cyan): varying ITD over a large range results in the rather symmetrical tuning function shown. Note that the only binaural change here is in the relative timing of these fixed PSPs. This is different for changes in IID only (ITD fixed at 0 ms), for which pairs of PSPs are shown in *Figure 8H* (IIDs of −20, 0, and +20 dB). As expected, the changes in level affect the amplitude of the PSPs, but they also have a large, clear effect on latency: the latency differences between onset of EPSP and IPSP are actually larger than the ITDs (±0.3 ms) imposed in *Figure 8I*. This results in a nonlinear interaction when both cues are combined, causing a marked functional change in the tuning function (*Figure 8J*). For the cue combination favoring the ipsilateral ear (both cues < 0; *Figure 8J*, left panel), the large and early EPSP is not effectively opposed by the small and later arriving IPSP: this results in a higher probability of spiking than for ITD or IID alone. For the combination favoring the contralateral ear (both cues > 0) (*Figure 8J*, right panel), a large and leading IPSP opposes a late and small EPSP: this results in a lower probability of spiking than for ITD alone. The effect of cue combination is therefore to remove the 'right' slope of ITD-tuning, and to generate a steep 'left' slope closer to 0 ITD (*Figure 8C*).

This is illustrated (*Figure 8K*, same cell) for a broad set of cue combinations. Artificial, single cue variations (*Figure 8H and I*) correspond to the vertical (gray) and horizontal (cyan) lines. For a real sound source moving in azimuth, the trajectory through this cue space is oblique (magenta): the exact trajectory depends on stimulus spectrum (*Maki and Furukawa, 2005*), but it generally courses from a region of high spike probability (lower left quadrant) to a region of low spike probability (upper right quadrant). Spike rates corresponding to these three cuts, for a broad range of cues, are shown in *Figure 8L*. Compared to the ITD-only condition (cyan), cue combination (magenta) indeed removes ambiguity by the absence of response for stimuli in the ipsilateral hemifield (IID >0, ITD >0), and results in a steeply-sloped tuning function positioned closer to 0. More limited datasets for three other cells, showing similar effects, are shown in *Figure 8—figure supplement 2*.

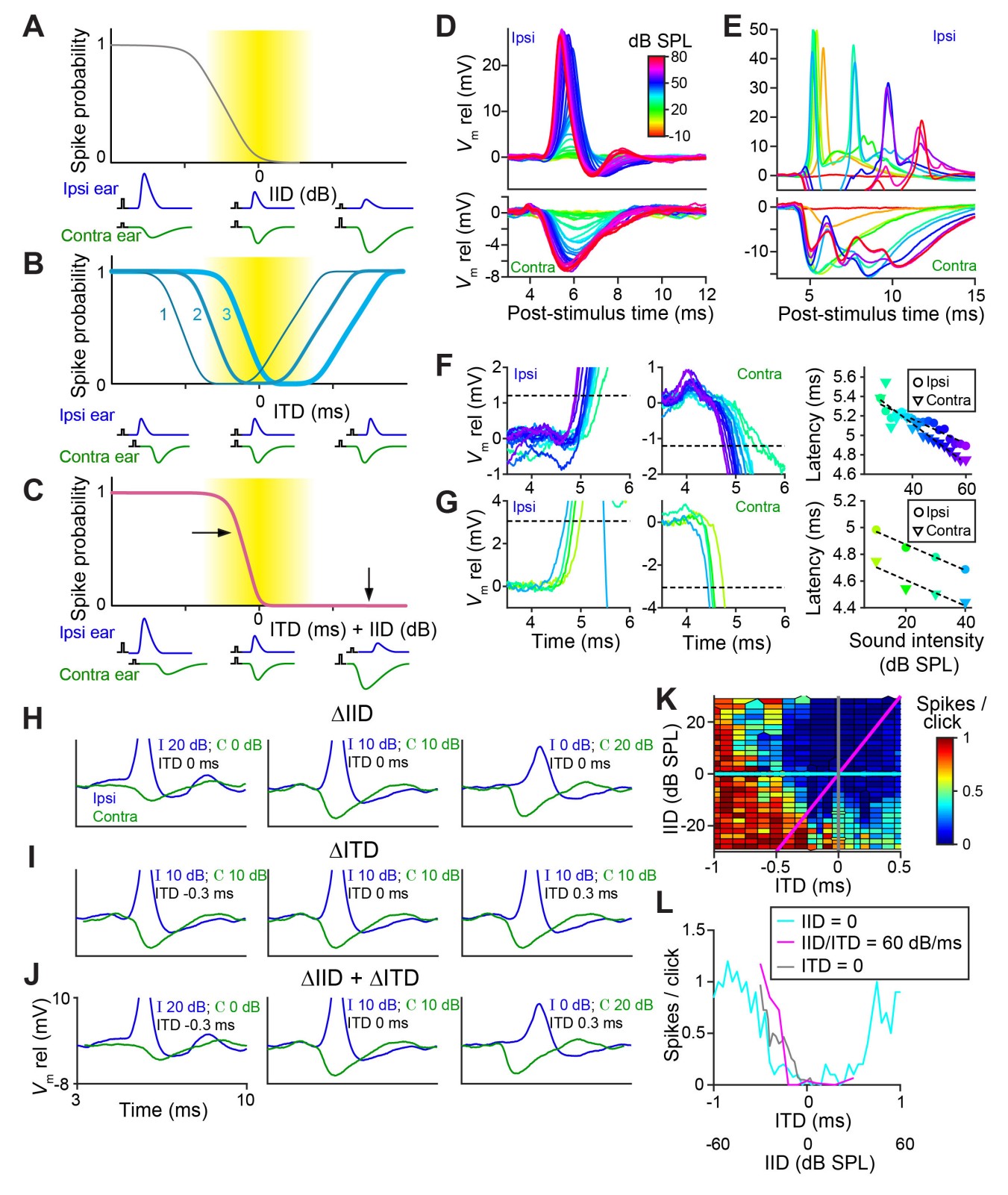

**Figure 8.** LSO neurons show graded latency-intensity changes which disambiguate spatial tuning. (**A**) Cartoon showing change in spike probability for changing IID. Yellow area shows approximate region of physiological IID values. Traces below each plot represent the timing and amplitude of ipsi- and contralateral synaptic events to click-pairs with different IID values. (**B**) Cartoon showing change in spike probability for changing ITD, for three cases with different centering of the trough. (**C**) Cartoon showing change in spike probability for combined changes in ITD and IID. Horizontal and

*Figure 8 continued on next page*

*Figure 8 continued*

vertical arrows indicate effects of adding IID to ITD: a rightward shift in the left slope, and inhibition of the right shoulder of the tuning function. (**D**) Average responses to ipsilateral clicks (top panel) and contralateral clicks (bottom panel) of different sound levels for a principal LSO cell (CF = 12 kHz; same cell as *Figure 2A*). (**E**) Similar to D, for a non-principal (marginal) LSO cell (CF = 4.1 kHz; same cell as *Figure 2B*). Colors as in D. (**F**) Ipsilateral (left panel) and contralateral responses (middle panel) for the same principal LSO cell as in D. Colors correspond to sound levels, as in D. Dashed lines indicate the threshold used to calculate latencies, a voltage difference relative to rest with an absolute value of 20% of the average IPSP amplitude (where IPSP amplitude was measured at the lowest sound level that led to the maximal spike rate when presented to the ipsilateral ear). For the ipsilateral ear, this voltage difference was depolarizing, for the contralateral ear, it was hyperpolarizing. Right panel: latency values as a function of sound level, corresponding to the data in the left and middle panel. (**G**) Similar to F, for the same non-principal LSO cell as in E. (**H**) Averaged monaural responses to click pairs with different IIDs, for a principal cell (CF = 3.5 kHz). (**I**) Similar to H, but now ITD varies and IID is kept constant at 0 dB. (**J**) Similar to H and I, but for combined changes in ITD and IID. (**K**) Voronoi diagrams of spike rate for different ITD and IID combinations, for the same principal cell as in H-J. Colors are clipped between the limits shown in the color scale. Gray and cyan lines connect data points of respectively IID and ITD functions. Diagonal magenta line connects data points for which there is a consistent change in ITD and IID (60 dB change in IID per 1 ms change in ITD, which is realistic for this CF [*Maki and Furukawa, 2005*]). Data was pooled across different sound levels. (**L**) Gray, cyan, and magenta functions show spike rates along the lines of the same color in K. Data from the same principal LSO cell as in H-K. Numerical data represented as graphs in this figure are available in a source data file (*Figure 8—source data 1*).

The online version of this article includes the following source data and figure supplement(s) for figure 8:

**Source data 1.** Excel table with data represented in this figure.
**Figure supplement 1.** Individual traces corresponding to the mean data shown in *Figure 8D and E*.
**Figure supplement 1—source data 1.** Excel table with data represented in this figure.
**Figure supplement 2.** Similar as *Figure 8K and L*, for three additional LSO neurons.
**Figure supplement 2—source data 2.** Excel table with data represented in this figure.

In summary, striking specializations at three levels combine to make LSO principal cells spatially tuned to transient sounds. Exquisite timing in afferent inputs supplies these neurons with temporally punctate events; the intrinsic properties of the neurons enable these events to interact at a sub-milli-second timescale; and the opposite sign and strategic location of the inputs enable input from one ear to veto the input from the other ear. The net result is sharp tuning to sound transients, which moreover is coherent with IID-tuning to sustained sounds in non-principal cells.

## Discussion

Our data lead to a new view of brainstem binaural processing, departing strongly from the previously accepted roles of the MSO as a timing comparator and the LSO as an intensity comparator. We find that both excel as timing comparators, be it for different types of sounds, complementary in frequency range and temporal characteristics. Our data show that principal LSO cells are significantly more temporally specialized than was previously appreciated, toward one specific, highly ecologically relevant form of binaural sensitivity which has received little attention: to sound transients (*Joris and Trussell, 2018*). Using diverse specializations, excitatory and inhibitory afferent circuits supply exquisitely timed PSPs to both MSO and LSO. By directing well-timed inhibition to the AIS, and combined with fast membrane properties of the LSO principal cells themselves, this circuit enables the output of one ear to veto the output of the other ear in a manner that is punctate in space and time. In contrast, binaural sensitivity of MSO neurons to these stimuli is surprisingly poor.

Traditionally, the LSO is viewed as the brainstem nucleus underlying behavioral sensitivity to IIDs. A long-standing problem with this depiction is that it lacks a rationale for the extreme features of the LSO-circuit, which hinder, rather than help IID sensitivity and which suggest a key role for timing. These features include large axosomatic synapses such as the calyx of Held, differential axon diameters on ipsi- and contralateral side, and fast membrane properties of monaural inputs (*Joris and Trussell, 2018*). Despite these features, ITD-sensitivity of LSO neurons is weak (*Caird and Klinke, 1983; Joris, 1996; Joris and Yin, 1995; Tollin and Yin, 2005; Tollin and Yin, 2002*), except to sound transients as documented in vivo for a limited number of neurons (*Caird and Klinke, 1983; Irvine et al., 2001; Joris and Yin, 1995; Park et al., 1996*) and in vitro with bilateral electrical shocks (*Sanes, 1990; Wu and Kelly, 1992*). It was recently argued that spatial sensitivity to high-frequency transients is particularly important for small mammals living near the ground plane, to enable detection of adventitious transient sounds generated by movement of nearby animals (*Joris and Trussell, 2018*), which provides a rationale for the presence of the calyx of Held and other temporal features

in the LSO-circuit. The data reported here are largely in line with this hypothesis. Combined with the recent finding that principal neurons of the LSO have fast kinetics that have been undersampled in extracellular studies (*Franken et al., 2018*), the data underscore that temporal aspects of binaural sensitivity are an essential feature of this nucleus.

LSO neurons show acute tuning to ITDs of transient stimuli to an extent that surpasses that of neurons in the MSO, which is classically regarded as the nexus of ITD-sensitivity (*Figures 1–3*). Intracellular traces to monaural stimulation reveal the presence of extraordinarily well-timed excitation and inhibition in LSO neurons (*Figure 2*). We discovered a 'prepotential' (*Figure 2A*, *Figure 2—figure supplement 1C*) preceding the IPSP with short latency in response to a transient at the contralateral ear, suggesting high synchronization between the many small inhibitory inputs. In response to binaural stimulation, inhibition is remarkable in its depth, temporal acuity, reliability, and limited duration of its effect. Effective interaction between EPSP and IPSP occurs over a time window which is only a small fraction of the latter's duration, and generates steeply-sloped and narrow ITD-tuning (*Figures 1* and *2*). Application of inhibition in vitro by somatic conductance clamp (*Figure 4*) or current injection (*Figure 4—figure supplement 1*), was ineffective to completely suppress spiking, as opposed to synaptically driven inhibition. This suggested that at least some synaptically evoked inhibition acts electrotonically closer to the spike initiation region in the axon. Indeed, morphological examination at the light (*Figure 5*) and EM (*Figure 6*) level revealed glycinergic terminals at the AIS of LSO but not MSO neurons (*Figure 6—figure supplement 1B*). Computational modeling demonstrated that inhibition targeting the soma and the AIS can suppress spiking more strongly than inhibition limited to the soma (*Figure 7*).

Our assertion is not that there is no IID-sensitivity in LSO, which has been abundantly demonstrated both to sustained and transient sounds, or that ITD is the only important binaural cue. Rather, it is that the LSO is temporally specialized toward binaural processing of sound transients. This is most easily explored with 'pure' temporal cues, that is ITDs of transients. However, these specializations will be engaged by, and affect responses to, any stimulus transient, even if stimulus ITD is not varied explicitly. Our recordings show directly that IIDs affect both amplitude and timing of EPSPs and IPSPs (*Figure 8*). It has often been proposed that IIDs are translated to ITDs through a peripheral latency mechanism (the 'latency hypothesis' [*Jeffress, 1948*]). Response latency generally decreases with sound level, so an acoustic IID would generate a neural ITD pointing to the same side. Human psychophysical studies do not support a simple IID-to-ITD conversion for low-frequency, ongoing sounds (*Domnitz and Colburn, 1977*). Indeed, for such sounds, IIDs are small (*Maki and Furukawa, 2005*), and the relationship between intensity and latency is complex (*Michelet et al., 2010*). However, EPSPs and IPSPs show large and systematic latency changes in response to transient sounds (*Figure 8D–G*). Physiological evidence for an interaction between IID and ITD has been observed for transient responses in a variety of species and anatomical structures (*Irvine et al., 2001*; *Joris and Yin, 1995*; *Park et al., 1996*; *Pollak, 1988*; *Yin et al., 1985*), but in these extracellular recordings, the underlying cellular mechanisms could not be assessed. Our intracellular recordings enabled direct examination and comparison of amplitude and timing of IPSPs and EPSPs and their relation to binaural responses. Our results suggest a different view of the role of latency changes. Both through the properties of its inputs and its intrinsic properties, the LSO is uniquely endowed to combine the two binaural cues. First, PSPs are not only extraordinarily precisely timed but also scale in both amplitude and latency with intensity: the large IIDs present at high frequencies (20 dB or more [*Maki and Furukawa, 2005*]) translate into delays that are substantial relative to the animal's headwidth (~120 μs for gerbil) and that add to the stimulus ITD. Second, the ears have opposite signs: one ear can veto the other ear but only over a very narrow time window. These properties, which rely on a range of specialized features both in the input pathway and the LSO cells themselves, all combine in neural space toward a single pair of PSPs that results in an unambiguous output signaling an ipsilateral (high output) or contralateral (no output) sound source for the range of cue values available to the animal (*Figure 8*).

We were able to collect data on exhaustive combinations of ITD and IID for only a small number of cells, since this requires holding the neurons very long. It is therefore not clear where the slopes of the tuning functions are positioned relative to the physiological range for the population of principal LSO neurons (*Figure 8K,L*; *Figure 8—figure supplement 2B*). We find that this can also vary with sound level (*Figure 1—figure supplement 3A,B*). Future studies need to address how these

cells respond to adventitious sounds in natural environments using stimuli in virtual space or free-field.

Comparison of monaural and binaural responses reveals why the EE (excitatory-excitatory) interaction underlying coincidence detection in MSO results in poorer binaural sensitivity to ITDs of clicks than the IE (inhibitory-excitatory) interaction underlying anti-coincidence detection in LSO. The multiplicative interaction in MSO hinges on subthreshold monaural inputs. Indeed, earlier modeling work has shown that a high level of monaural coincidences reduces sensitivity to ITDs for sustained sounds (*Colburn et al., 1990*; *Franken et al., 2014*), and various mechanisms counter the presence of monaural coincidences (*Agmon-Snir et al., 1998*). However, transient stimuli can synchronize sufficient monaural inputs to cause suprathreshold monaural coincidences in MSO neurons, and the sign of these responses is positive (increased spike rate) and identical for monaural stimulation (of either ear) and binaural stimulation, so that little response increment is gained with binaural stimulation. In contrast, in LSO neurons the response sign is opposite for the two ears and maximal binaural interaction is obtained when ITD causes an alternation in sign from robust excitation to profound inhibition. In this subtractive mechanism, strong monaural responses (of opposite sign) yield maximal binaural interaction.

The role of axo-axonic synapses in these temporal computations directly links axonal inhibition to a clear physiological operation (extraction of binaural cues toward sound localization). This makes this circuit a model system to study how such synapses modulate neuronal output at the single-cell level. Indeed, despite the well-known occurrence of axo-axonic synapses provided by chandelier cells in cortex, their physiological role has to a large extent remained a mystery (*Pan-Vazquez et al., 2020*). Our data indicate that inhibitory axo-axonic synapses cooperate with somatic inhibition to drastically reduce neuronal output with high temporal precision. This may help shed light on the function of such synapses elsewhere in the brain.

We conclude that the LSO pathway is not a simple IID pathway but consists of at least two subsystems with a coherent code for sound laterality. Principal cells encode spatial location of sound transients based on exquisite temporal sensitivity; non-principal cells encode spatial location of ongoing sound features, based on intensity. Given that principal cells are the most numerous cell type (*Helfert and Schwartz, 1987*; *Saint Marie et al., 1989*), and given the extreme nature of specializations in the afferent pathway and how they interact via the principal cells, the time-based role of LSO is the more dominant (but previously underestimated) role. This role may support fast and reliable localization of adventitious sounds, such as those made by approaching predators or prey.

# Materials and methods

## Key resources table

| Reagent type (species) or resource | Designation | Source or reference | Identifiers | Additional information |
|---|---|---|---|---|
| Strain, strain background (*Meriones unguiculatus*, male and female) | Wildtype | In vivo: Janvier Labs In vitro: Animals raised in colony at at UT-Austin Animal Resource Center (breeders obtained from Charles River Laboratories) | | |
| Antibody | Anti-Synaptophysin1 (Guinea pig polyclonal) | Synaptic Systems | Cat# 101–004; RRID:AB_ 1210382 | (1:500) |
| Antibody | Anti-AnkyrinG (rabbit polyclonal) | *Galiano et al., 2012* | | (1:200) Courtesy of Dr. Matthew Rasband (Baylor College of Medicine) |
| Antibody | Anti-Gephyrin (mouse monoclonal) | Synaptic Systems | Cat# 147–011; RRID:AB_ 887717 | (1:200) |
| Software, algorithm | MATLAB | The Mathworks | | |
| Software, algorithm | IGOR-Pro | Wavemetrics | | |

*Continued on next page*

*Continued*

| Reagent type (species) or resource | Designation | Source or reference | Identifiers | Additional information |
|---|---|---|---|---|
| Software, algorithm | MafDC | *Yang et al., 2015* | | Courtesy of Dr. Matthew Xu-Friedman |
| Software, algorithm | SIM-post processing software | Zeiss | | |
| Software, algorithm | Metamorph | Molecular Devices | | |

## Animals

For the in vivo recordings, adult (P60-P90) and juvenile (range P22-P35, median P29) Mongolian gerbils (*Meriones unguiculates*) of both sexes were used. The animals had no prior experimental history and were housed with up to six per cage. This study was performed in accordance with the recommendations in the Guide for the Care and Use of Laboratory Animals of the National Institutes of Health. All in vivo procedures were approved by the KU Leuven Ethics Committee for Animal Experiments (protocol numbers P155/2008, P123/2010, P167/2012, P123/2013, P005/2014). After perfusion, the tissue of some of these animals were used for electron microscopic analysis.

For the in vitro recordings, Mongolian gerbils aged P19-22 were used. For the immunohistochemistry experiments, Mongolian gerbils aged P24-26 were used. All in vitro recording and immunohistochemistry experiments were approved by the University of Texas at Austin Animal Care and Use Committee in compliance with the recommendations of the United States National Institutes of Health.

The methods for in vivo and in vitro patch clamp recording and electron microscopy have been previously described (*Franken et al., 2018*; *Franken et al., 2016*; *Franken et al., 2015*) and are briefly summarized here.

## Surgery for in vivo electrophysiology

The animals were anesthetized by an intraperitoneal injection of a mixture of ketamine (80–120 mg/kg) and xylazine (8–10 mg/kg) in 0.9% NaCl. Anesthesia was maintained by additional intramuscular injections of a mixture of ketamine (30–60 mg/kg) and diazepam (0.8–1.5 mg/kg) in water, guided by the toe pinch reflex. Body temperature was kept at 37°C using a homeothermic blanket (Harvard Apparatus, Holliston, MA, USA) and a heating lamp. The ventrolateral brainstem was exposed by performing a transbulla craniotomy. This access allowed us to record from either LSO or MSO neurons. The contralateral bulla was opened as well to maintain acoustic symmetry. Meningeal layers overlying the exposed brainstem were removed prior to electrode penetration, and cerebrospinal fluid (CSF) leakage wicked up or aspirated. Pinna folds overlying the external acoustic meatus were removed bilaterally to ensure proper delivery of the acoustic stimuli.

## In vivo electrophysiology

Patch clamp pipettes were pulled from borosilicate capillaries (1B120F-4, World Precision Instruments, Inc, Sarasota, FL, USA) with a horizontal puller (P-87, Sutter Instrument Co., Novato, CA, USA). When filled with internal solution, electrode resistance was 5–7 MΩ, measured in CSF. The internal solution contained (in mM) 115 K-gluconate (Sigma); 4.42 KCl (Fisher); 10 $Na_2$ phosphocreatine (Sigma); 10 HEPES (Sigma); 0.5 EGTA (Sigma); 4 Mg-ATP (Sigma); 0.3 Na-GTP (Sigma); and 0.1–0.2% biocytin (Invitrogen). pH was adjusted to 7.30 with KOH (Sigma) and osmolality to 300 mOsm/kg with sucrose (Sigma). A patch clamp amplifier (BC-700A; Dagan, Minneapolis, MN, USA) was used to obtain membrane potential recordings, where the analog signal was low-pass filtered (cut-off frequency 5 kHz) and digitized at 50–100 kHz (ITC-18, HEKA, Ludwigshafen/Rhein, Germany; RX8, Tucker-Davis Technologies, Alachua, FL, USA). Data was collected using MATLAB (The Mathworks, Natick, MA, USA). In vivo whole-cell recordings were obtained from LSO and MSO neurons. Neurons were identified as principal LSO neurons, non-principal LSO neurons or MSO neurons using the same morphological and/or physiological criteria as in our earlier work (*Franken et al., 2018*; *Franken et al., 2015*). LSO and MSO samples cover the same range of CFs (range MSO: 508 Hz - 8000 Hz; range LSO: 437 Hz – 12021 Hz) (see e.g. *Figure 1K and L*). Series resistance was

61.8 ± 3.77 MΩ (mean ± s.e.m., 19 cells, leaving out one cell with series resistance >100 MΩ) for LSO neurons and 70.6 ± 2.98 MΩ (mean ± s.e.m., 28 cells, leaving out two cells with series resistance >100 MΩ) for MSO neurons. Opening resting membrane potential was −56.9 ± 0.56 mV (mean ± s.e.m., 20 cells) for LSO neurons and −53.6 ± 0.74 mV (mean ± s.e.m., 27 cells) for MSO neurons, both corrected for a 10 mV liquid junction potential. Reported membrane potentials are typically presented as $V_m$ rel, that is after subtracting resting membrane potential.

## Acoustic stimuli

In vivo recordings were done with the animal in a double-walled sound-proof booth (IAC, Niederkrüchten, Germany). TDT System II hardware (Tucker-Davis Technologies, Alachua, FL, USA) was used to generate and present sound stimuli, using MATLAB (The Mathworks, Natick, MA, USA). Acoustic speakers (Etymotic Research Inc, Elk Grove Village, IL, USA) attached to hollow ear bars were positioned over the external acoustic meatus bilaterally. Acoustic calibration was done before each recording using a probe microphone (Bruel and Kjaer, Nærum, Denmark). Characteristic frequency (CF) was measured with a threshold-tracking algorithm during ipsilateral short tone presentation, using either spikes or large EPSPs as triggers. CF was defined as the tone frequency with the lowest threshold. For some cells, CF was not recorded: we then report best frequency (BF) that is the frequency that elicits the maximal response for tones of the same sound level.

Responses were obtained to monaural and binaural rarefaction clicks (rectangular pulse scaled according to the acoustic calibration, duration 20 μs). Monaural ipsilateral and monaural contralateral responses were typically obtained to different sound levels (from −10 or 0 dB SPL to ~80 dB SPL, in steps of 2–10 dB), for 5 or 10 repetitions per sound level and 100 or 200 ms in between successive clicks. A binaural data set was often obtained using the same parameters. Then, responses were obtained to binaural clicks where ITD was varied. Sound level was set at a value for which monaural ipsilateral responses were suprathreshold. ITD was varied in steps of 100 μs, and 10–20 repetitions were typically obtained per ITD. ITD responses were often obtained at several sound levels. For some neurons, responses were obtained to binaural clicks for which IID was varied. Positive ITDs and positive IIDs refer to stimuli for which respectively the contralateral stimulus leads the ipsilateral stimulus or the contralateral stimulus is more intense than the ipsilateral stimulus.

For many of these neurons, responses to tonal stimuli have been reported before (LSO: *Franken et al., 2018*; MSO: *Franken et al., 2015*).

## Analysis

Data analysis was performed in MATLAB. Custom MATLAB code for the computational model is provided as a source code file.

## Analysis of in vivo data

ITD functions were smoothed by convolution with a three-point Hanning window (MATLAB function *hanning*) for the population plots in *Figure 1G–I* and *Figure 1—figure supplement 2*, and before measuring slope steepness (*Figure 1J*) and halfwidth (*Figure 1K*).

To quantify the modulation of spike rate as a function of ITD (*Figure 1L*), we used the ITD-SNR metric which has been described by *Hancock et al., 2010*. ITD-SNR was calculated for all data sets with ≥10 repetitions and is defined as

$$ITD \, SNR \, = \, \frac{\sigma_{ITD}^2}{\sigma_{tot}^2}$$

$\sigma_{ITD}^2$ stands for the variance in spike counts related to the ITD and is defined as

$$\sigma_{ITD}^2 = \frac{1}{N_s} \sum_{s=1}^{N_s} \left( \bar{R}_s - \bar{R} \right)^2$$

where $\bar{R}_s$ is the mean spike count for each ITD value $s$ across trials and $\bar{R}$ is the grand mean of $\bar{R}_s$ across ITD values. $\sigma_{tot}^2$, the total variance is defined as

$$\sigma^2_{tot} = \frac{1}{N_S N_T} \sum_{s=1}^{N_s} \sum_{t=1}^{N_T} \left( R_{s,t} - \bar{R} \right)^2$$

where $N_s$ is the number of different ITD values, $N_T$ is the number of trials per ITD and $R_{s,t}$ is the spike count in response to ITD $s$ during trial $t$.

To compare binaural responses to monaural response, we defined a summation ratio. For LSO neurons, this ratio is defined as

$$SR_{LSO} = median \left( \frac{\bar{R}_{spl,ipsi} + \bar{R}_{spl,contra}}{\bar{R}_{spl,bin}} \right)$$

where $\bar{R}_{spl}$ stands for the mean spike count across trials to a stimulus with sound level $spl$. $SR_{LSO}$ is then calculated as the median ratio across sound levels. Because LSO neurons are excited by ipsilateral sounds but inhibited by contralateral sounds, a strong binaural effect means that the response to binaural stimuli is a lot smaller than the sum of monaural ipsilateral and monaural contralateral responses, and this will result in large values of $SR_{LSO}$. If instead binaural stimulus presentation results in the same average spike count as the sum of monaural ipsilateral and monaural contralateral stimulation, $SR_{LSO}$ will be equal to 1.

For MSO neurons, the summation ratio is defined instead as

$$SR_{MSO} = median \left( \frac{\bar{R}_{spl,bin}}{\bar{R}_{spl,ipsi} + \bar{R}_{spl,contra}} \right)$$

Since MSO neurons are excited by monaural ipsilateral as well as monaural contralateral sounds, a strong binaural effect will result in $\bar{R}_{spl,bin}$ being much larger than the sum of $\bar{R}_{spl,ipsi}$ and $\bar{R}_{spl,contra}$. Inverting the ratio in the definition of $SR_{MSO}$ compared to $SR_{LSO}$ thus means that both metrics are >> 1 when there is a significant binaural advantage compared to monaural stimulation.

To generate the Voronoi diagrams (*Figure 8K*; *Figure 8—figure supplement 2*), we used the MATLAB function *voronoin* to generate Voronoi cells for all available combinations of ITD and IID. ITD values were divided by 0.05 ms/dB before feeding them in *voronoin* together with IID values in dB. Each Voronoi cell was colored according to spike rate.

## In vitro electrophysiology

The animals were perfused and subsequently sectioned under a Na+-free solution containing: 135 mM N-Methy-D-Glucamine (NMDG), 20 mM D-Glucose, 1.25 mM KCl, 1.25 mM KH2PO4, 2.5 mM MgSO4, 0.5 mM CaCl2, and 20 mM Choline Bicarbonate (pH adjusted to 7.45 using NMDG powder, final osmolarity: 310 mOsm). Coronal slices were prepared and incubated at 37°C in a recovery solution: 110 mM NaCl, 25 mM D-Glucose, 2.5 mM KCl, 25 mM NaHCO3, 1.25 mM NaH2PO4, 1.5 mM MgSO4, 1.5 mM CaCl2, 5 mM N-Acetyl-L-Cystine, 5 mM Sodium ascorbate, 3 mM sodium pyruvate, and 2 mM Thiourea (pH adjusted to 7.45 with NaOH, final osmolarity: 310 mOsm). Following 30–45 min of recovery, slices were maintained at room temperature for >30 min before recording.

Whole-cell current-clamp recordings were made using Dagan BVC-700A amplifiers. Voltage data was filtered at 5 kHz, digitized at 100 kHz, and stored on computer using Igor Pro (Wavemetrics). Recording electrodes were pulled from borosilicate glass (1.5 mm OD; 4–8 MΩ) and filled with intracellular solution containing 115 mM K-gluconate, 4.42 mM KCl, 0.5 mM EGTA, 10 mM HEPES, 10 mM, Na2Phosphocreatine, 4 mM MgATP, and 0.3 mM NaGTP, osmolality adjusted to 300 mOsm/L with sucrose, pH adjusted to 7.30 with KOH. All recordings were carried out at 35°C with oxygenated ACSF perfused at a rate of ~2–4 mL/min, and bridge balance and capacitance compensation were monitored throughout. Series resistance was maintained below 10 MΩ for dynamic clamp experiments, and electrode capacitance was fully compensated. All membrane potentials shown are corrected for a 10 mV liquid junction potential.

ITD dynamic clamp experiments were carried out and analyzed under control of a user interface using Igor Pro routines kindly provided by Dr. Matthew Xu-Friedman (MafDC; *Yang et al., 2015*). Dynamic clamp was implemented at 50 kHz via an ITC-18 computer interface (Heka Instruments).

Excitatory and inhibitory conductances and currents were simulated with double exponential waveforms (EPSCs/EPSGs: time constants = 0.1 ms rise, 0.18 ms decay, reversal potential of 0 mV; IPSCs/IPSGs: time constants = 0.45 ms rise, 2.0 ms decay, reversal potential of −75 mV), based on published literature for the age range used for slice experiments in gerbils and other rodents (*Walcher et al., 2011*; *Kullmann and Kandler, 2001*; *Balakrishnan et al., 2003*; *Kakazu et al., 1999*; *Ehrlich et al., 1999*). The amplitude of EPSCs and EPSGs was adjusted 20% above threshold for reliable spike initiation. The peak conductance of IPSGs and amplitude of IPSCs were adjusted so that an individual event elicited a 5–10 mV hyperpolarization from the resting potential (similar as IPSP amplitudes observed in vivo at a similar resting potential). The resting potential was maintained at −60 mV with direct current through the recording electrode to maintain consistent driving forces on excitatory and inhibitory synaptic currents across different experiments. Synaptic stimuli were evoked through glass pipettes (50–100 μm dia.) via a constant current stimulator (Digitimer DS3), and presented with random temporal offset intervals. Small current steps were interleaved to monitor input resistance. Synaptic stimulation was ipsilateral to the LSO for excitatory input stimulation, or near the center of the MNTB for inhibitory stimulation. Excitatory and inhibitory responses were isolated through the inclusion of 1 μM strychnine or 10 μM NBQX to the bath, respectively. Stimulation intensity was also adjusted so that action potential probability at optimal synaptic timing was close to, but less than 100%, to avoid saturation.

Similar to the in vivo ITD functions, slope values of in vitro functions (*Figure 4E*, *Figure 4—figure supplement 1C*) were measured after smoothing the function with a three-point Hanning window (MATLAB function *hanning*).

## Immunohistochemistry and SIM microscopy

The brainstem of Mongolian gerbils (P24-26) were rapidly dissected, blocked in the coronal plane, and drop fixed in cold 4% paraformaldehyde for 30–60 min. Tissue was cryoprotected in a gradient of sucrose solutions (20% sucrose overnight, 30% sucrose overnight; 4°C), and subsequently embedded in Optimal Cutting Temperature (O.C.T.) media. Sections were sliced on a cryostat (16–20 μm; −19°C) and mounted on slides for immunohistochemistry.

Tissue on slides were rehydrated in 0.1M PBS for 5–10 min. Sections were blocked and permeabilized with PBTGS (10% Goat Serum, 0.3% Triton in 0.1M PBS) for 1.5 hr in a humidity chamber at room temperature on a slow-moving shaker. The tissue was then incubated with a primary antibody solution in PBTGS for 48 hr at 4°C. Primaries included mouse anti-Gephyrin (1:200; Synaptic Systems [cat. #147–011]), rabbit anti-AnkyrinG (1:200; Courtesy of Dr. Matthew Rasband; Baylor College of Medicine [*Galiano et al., 2012*]), and guinea pig anti-Synaptophysin1 (1:500, Synaptic Systems [cat. #101–004]). After primary incubation, the tissue was gently washed 3x with 0.1M PBS (5;10;15 min intervals) at room temperature. Tissue was then incubated for 2 hr at room temperature in a PBTGS secondary antibody solution including goat anti-mouse Alexa568 (1:200; Abcam [ab175473]), goat anti-rabbit Cy2 (1:200; Jackson Laboratories Inc, [111-225-144]), and goat anti-guinea pig 647 (1:200, Abcam [ab150187]). Slides were again washed 2x with 0.1M PBS, and a third wash (15 min) was done in 0.05M PBS. Tissue was then partially dried, and cover slipped with Fluoromount-G containing DAPI. After drying for 24 hr, slides were cover slipped and sealed with nail polish (24 hr) for imaging.

Low power images of the LSO (×112 magnification) were taken on a Zeiss Stereoscope (Axio Zoom.V16). LSO nuclei were subsequently imaged using SIM-microscopy (Zeiss LSM710 with Elyra S.1) and z-stacks of targeted regions were generated (~0.5 μm optical sections; total ~15 μm). Post-SIM processing of multichannel images (488; 568; 647 nm) was done offline using Zeiss SIM post-processing software with a Wiener filter setting between −5.0 and −5.2, followed by individual channel deconvolution using Metamorph software (Molecular Devices). The resulting images were not used for direct quantifications of anatomical structures due to the presence of some remaining SIM processing artifacts introduced by unavoidable light scatter in the tissue sections. Imaging was targeted toward AIS that ran in a single optical plane.

## Histology and electron microscopy of cells labeled with biocytin during in vivo recording

After the recording session, the animal was euthanized with pentobarbital and perfused through the heart with 0.9% NaCl followed by paraformaldehyde (PFA) 4% in 0.1M phosphate buffer or (for electron microscopy analysis) by PFA 1%/glutaraldehyde 1% and PFA 2%/glutaraldehyde 1%. Tissue processing methods for light and electron microscopy have been described previously (*Franken et al., 2018*; *Smith et al., 2010*; *Smith et al., 2005*) and are summarized here. The brain was dissected out of the skull and stored in PFA 1%/glutaraldehyde 1% for at least 24 hr. A vibratome was used to cut sections (70 μm thick) and the DAB-nickel/cobalt intensification method (*Adams, 1981*) was then used to visualize biocytin. After rinsing in phosphate buffer, free-floating sections were inspected with a light microscope to locate the labeled neuron. Sections containing the labeled cell body and relevant portions of the axon were processed for electron microscopy. These were fixed in 0.5% osmium tetroxide for 30 min, rinsed and dehydrated through a series of graded alcohols and propylene oxide. They were then placed in unaccelerated Epon-Araldite resin and transferred into a fresh batch of unaccelerated resin overnight. The sections were then embedded in plastic and flat mounted in accelerated resin between Aklar sheets at 65°C. The region of the embedded sections containing the labeled neuron and its axon, that typically arose from the cell body, were cut out and mounted on the flattened face of a plastic beam capsule. The 70-μm-section was re-sectioned into 3 μm sections which were placed on a glass coverslip. The 3 μm sections containing the labeled neuron and the first 50–100 μm of the axon were selected and remounted on a beam capsule. Thin sections (70–80 nm) were then cut and mounted on coated nickel grids. These sections were stained with uranyl acetate and lead citrate and examined using a Philips CM-120 electron microscope.

Cell types were identified using morphological and physiological criteria as described before (*Franken et al., 2018*; *Helfert and Schwartz, 1987*). Briefly, principal LSO cells were identified by the central location of their cell body in the LSO, bipolar dendritic arbors in the transverse plane, high levels of cell body synaptic coverage at the E.M. level (>50%), small action potentials and fast subthreshold kinetics.

## Statistics

All error bars represent standard error of the mean. Data distribution was not formally tested for normality. Exact p-values are given and all p values are two-tailed. Statistical significance was defined as $p < 0.05$. A non-parametric effect size measure, $\theta$, estimated as the Mann-Whitney $U$ statistic divided by the product of sample sizes, is reported for two-sample statistical analyses (*Newcombe, 2006a*; $\theta$ ranges from 0 to 1 and $\theta = 0.5$ in case of no effect). 95% confidence intervals for $\theta$ were calculated using freely available software in Excel developed by Dr. Robert Newcombe (Cardiff University, http://profrobertnewcomberesources.yolasite.com/, using method five by *Newcombe, 2006b*). Pearson's correlation coefficient $r$, and associated 95% confidence interval and p-value were calculated using the MATLAB function *corrcoef*. For in vivo data, multiple data sets were often available from the same cell (at different sound intensities, and/or at different frequencies [for tones]). Before doing statistical analyses, metrics were averaged per cell across these different data sets so that each cell contributes one data point to the analysis.

## Computational model

### Two-compartment model

A two-compartment model gives a minimal description of a cell with two distinguishable spatial regions. The two regions in the model are soma (compartment 1) and axon initial segment region (AIS, compartment 2). Voltage dynamics in the two compartments are governed by the differential equations:

$$AC_mV_1^{'} = -AG_{lk1}(V_1 - E_{lk,1}) - g_{ax}(V_1 - V_2) - I_{syn,1}$$

$$\alpha AC_mV_2^{'} = -\alpha AG_{lk2}(V_2 - E_{lk,2}) - g_{ax}(V_2 - V_1) - I_{ion} - I_{syn,2}$$

Specific membrane capacitance is $C_m$ = 0.9 μF/cm2 (*Gentet et al., 2000*). The ratio of membrane

areas between the two compartments is α = 0.12, based on anatomical observations that the soma is ellipsoid in shape with length 20.4 µm and width 9.5 µm (*Helfert and Schwartz, 1987*) and the AIS is cylindrical with length 20 µm and diameter 1 µm (*Bender and Trussell, 2012*). The remaining parameters are the effective soma area (A), leak reversal potential ($E_{lk}$), leak conductance density ($G_{lk}$, can differ in the two compartments), and axial conductance ($g_{ax}$). We describe these parameters in more details below. Their values are set so that passive dynamics of $V_1$ match typical properties LSO neurons (resting potential −60 mV, input resistance 40 MΩ, and time constant 1 ms, [*Sanes, 1990*; *Ashida et al., 2017*]).

## Conductance parameters determined from steady-state responses

Conductance parameters are set based on passive, steady-state responses to constant inputs (following the approach of *Goldwyn et al., 2019*). The passive version of the two-compartment model is

$$c_m U_1^{'} = -g_1 U_1 - g_{ax}(U_1 - U_2) - I_1$$

$$\alpha c_m U_2^{'} = -g_2 U_2 - g_{ax}(U_2 - U_1) - I_2$$

where $U_1$ and $U_2$ measure the deviation of voltages from resting potential in each compartment, $c_m$ is the membrane capacitance (pF), $g_1$ and $g_2$ are passive (leak) conductance in each compartment (nS), and $I_1$ and $I_2$ are input currents (pA).

Steady-state voltages in response to constant inputs satisfy the linear equations

$$0 = -g_1 U_1 - g_{ax}(U_1 - U_2) - I_1$$

$$0 = -g_2 U_2 - g_{ax}(U_2 - U_1) - I_2$$

from which we calculate the input resistances for the two compartments:

$$R_1 = \frac{g_2 + g_{ax}}{(g_1 + g_{ax})(g_2 + g_{ax}) - g_{ax}^2}$$

$$R_2 = \frac{g_1 + g_{ax}}{(g_1 + g_{ax})(g_2 + g_{ax}) - g_{ax}^2}$$

and the transfer resistance between compartments

$$R_{12} = R_{21} = \frac{g_{ax}}{(g_1 + g_{ax})(g_2 + g_{ax}) - g_{ax}^2}.$$

From steady-state solutions to these equations, we can also measure the attenuation of voltage from one compartment to the other. Forward attenuation (soma-to-AIS) for constant input to the soma is

$$\kappa_{1 \to 2} \equiv \frac{U_2}{U_1} = \frac{g_{ax}}{g_{ax} + g_2}$$

and backward attenuation (AIS-to-soma) for constant input to the AIS is

$$\kappa_{2 \to 1} \equiv \frac{U_1}{U_2} = \frac{g_{ax}}{g_{ax} + g_1}$$

We refer to these as forward and background coupling constants (*Goldwyn et al., 2019*). They take values between 0 (no coupling, complete voltage attenuation) and 1 (complete coupling, no voltage attenuation). We invert the above relations to construct a family of models, uniquely determined by the pair of coupling constants (see also *Goldwyn et al., 2019*):

$$g_{ax} = \frac{1}{R_1}\left(\frac{\kappa_{2 \to 1}}{1 - \kappa_{1 \to 2}\kappa_{2 \to 1}}\right)$$

$$g_1 = g_{ax} \left( \frac{1}{\kappa_{2 \to 1}} - 1 \right)$$

$$g_2 = g_{ax} \left( \frac{1}{\kappa_{1 \to 2}} - 1 \right)$$

where the soma input resistance $R_1$= 40 MΩ (**Sanes, 1990**; **Ashida et al., 2017**). Values of these conductance parameters across the space of coupling constants are shown in **Figure 7—figure supplement 1A–C**.

A relation that we make use of in our analysis of AIS-targeting inhibition is

$$R_1 \kappa_{1 \to 2} = R_2 \kappa_{2 \to 1}.$$

This can be observed from the equations for input resistance and coupling constants, or by recognizing that $R_1 \kappa_{1 \to 2}$ are $R_2 \kappa_{2 \to 1}$ identical because they are both equivalent to the transfer resistance between compartments. Under a steady-state approximation, soma-targeting inputs affect AIS voltage in proportion to $R_1 \kappa_{1 \to 2}$ and AIS-targeting inputs affect AIS voltage in proportion to $R_2$. The ratio of these is $\kappa_{2 \to 1} = R_1 \kappa_{1 \to 2} / R_2$ and thus the backward coupling constant plays a key role in amplifying the strength of AIS-targeting inhibition. We used $\kappa_{2 \to 1}$=0.6 in most simulations (**Figure 7A,B**) because action potential amplitudes in the soma are approximately 30 mV for this backward coupling strength, consistent with AP sizes observed in vivo (**Franken et al., 2018**) and confirmed that our results remained qualitatively consistent for other backward coupling strengths (**Figure 7C**).

### Effective soma area determined by passive dynamics

Soma dynamics are commonly described by a membrane time constant describing the rate of exponential decay to rest. In contrast, voltage in the passive two-compartment model evolves on two time scales. If the area ratio between soma and AIS ($\alpha$) is very small, a separation of time-scales argument can be used to isolate a dominant time scale (**Goldwyn et al., 2019**). We could not pursue this approach here since $\alpha = 0.12$ is not sufficiently small. Nonetheless, we found it possible to roughly match the $U_1$ dynamics in the passive two-compartment model to exponential decay with a single time constant.

To do this, we considered decay from a steady-state holding potential in the passive model:

$$c_1 U_1' = -g_1 U_1 - g_{ax}(U_1 - U_2)$$

$$\alpha c_1 U_2' = -g_2 U_2 - g_{ax}(U_2 - U_1)$$

$$U_1(0) = 1, U_2(0) = \kappa_{1 \to 2}$$

Or, more compactly in matrix-vector notation as

$$\mathbf{u}' = -\mathbf{M}\mathbf{u}$$

where

$$\mathbf{M} = \begin{bmatrix} (g_1 + g_{ax})/c_1 & -g_{ax}/c_1 \\ -g_{ax}/\alpha c_1 & (g_2 + g_{ax})/\alpha c_1 \end{bmatrix}$$

and

$$\mathbf{u} = \begin{bmatrix} U_1(t) \\ U_2(t) \end{bmatrix}$$

and the initial value is $\mathrm{u}(0) = \begin{bmatrix} 1 \\ \kappa_{1 \to 2} \end{bmatrix}$.

Denoting the eigenvectors of **M** as $\mathbf{w_i}$ (i=1,2) with corresponding eigenvalues $\lambda_i$, the vector of voltage decay in both compartments is

$$\boldsymbol{u}(t) = a_1\boldsymbol{w}_1 e^{-\lambda_1 t} + a_2\boldsymbol{w}_2 e^{-\lambda_2 t}$$

where $a_1$ and $a_2$ are the coordinates of the initial value vector $\boldsymbol{u}(0)$ in the basis of eigenvectors (**Boyce et al., 2018**). We then compared the soma voltage $U_1(t)$ to exponential decay on a single time scale using the error function:

$$E(c_1) = \int_0^T \left( e^{-t/\tau_m} - U_1(t) \right)^2 dt.$$

where the membrane time constant is $\tau_{\mathrm{m}} = 1$ ms (**Sanes, 1990**; **Ashida et al., 2017**). With $\alpha = 0.12$ and passive conductance parameters determined by the choice of coupling constants, the only remaining free parameter in the formula for $U_1(t)$ is $c_1$. We determined a suitable value for $c_1$ by evaluating the error integral (with a finite upper limit, we use $T = 10$, but any limit sufficiently larger than $\tau_m$ gives similar results) and numerically determined the $c_1$ value that minimized the error function. **Figure 7—figure supplement 1D** shows values of $c_1$ across the space of coupling configurations.

To summarize the parameterization method to this point: known, typical anatomical and physiological measurements of LSO neurons define a family of passive models. Each member of this family differs in its forward and backward coupling constants but has nearly identical passive soma voltage dynamics (see **Figure 7—figure supplement 1G** for examples).

## Spike-generating currents

Spike-generating sodium current and high-threshold potassium current are located in the AIS region (compartment 2). Hodgkin-Huxley-type descriptions of these currents were adapted from previous LSO models (**Ashida et al., 2017**; **Wang and Colburn, 2012**), specifically the adjusted Wang-Colburn model in **Ashida et al., 2017**. For each coupling configuration, maximal sodium conductance was set so that spike threshold occurred for EPSPs in the soma of about 10 mV, similar to what we observed in vivo. We then set maximal high-threshold potassium conductance to be 10% of the maximal sodium conductance, consistent with **Ashida et al., 2017**. Values of maximal sodium conductance ($g_{Na}$) across the space of coupling constants are shown in **Figure 7—figure supplement 1E**. Examples of spiking dynamics for two different coupling configurations are shown in **Figure 7—figure supplement 1H**.

## Synaptic inputs

Synaptic currents were described as sums of unitary inputs of the form $g(t)(V_i - E_{syn})$ where g(t) is the conductance waveform, $V_i$ is the voltage in a compartment (excitation in soma only, inhibition can be in either soma or AIS), and $E_{syn}$ is reversal potential (0 mV for excitation and -75 mV for inhibition). Conductance waveforms are the same as used in our in vitro experiments: double exponentials with rise time 0.1 ms and decay time 0.18 ms for excitation, and rise time 0.45 ms and decay time 2 ms for inhibition. Maximal conductances were 2.3 nS for excitation and 3.1 nS for inhibition. Synaptic populations consisted of 20 excitatory inputs and 8 inhibitory inputs, consistent with previous models of LSO neurons in which inhibitory inputs were fewer but stronger than excitatory inputs (**Ashida et al., 2017**; **Gjoni et al., 2018b**). To explore the impact of AIS-targeting inhibition, simulations were performed with different arrangements of inhibitory inputs including all eight inhibitory inputs to the soma, six soma inputs and two AIS inputs, and only two AIS inputs.

Each individual synapse in the population is activated independently at a fixed time and with identical probability. Thus, the synaptic current of a given type (excitatory or inhibitory) is

$$I_{syn} = b_{N,p} g(t)(V_i - E_{syn})$$

where $b_{N,p}$ is binomial-distributed with success probability p = 0.84 (excitatory) and p = 0.92 (inhibitory) and maximum possible number of inputs (N = 20 for excitation, N =2,6, or 8 for inhibition depending on the arrangement of inhibitory inputs and compartment). These high probabilities reflect the fact that clicks elicit robust and reliable post-synaptic potentials in LSO neurons. The inhibitory probability of 0.92 was selected specifically so that the mean and variance of simulated IPSPs matched these statistics of IPSPs measured in a principal LSO cell in vivo in responses to contralateral clicks (**Figure 2A**, mean IPSP = -6.35 mV, variance 0.184 mV).

We repeated the computational study using conductance waveforms adapted from a recent in vivo study of mature LSO neurons (*Beiderbeck et al., 2018*). These conductance waveforms were also double exponentials but with substantially faster inhibition kinetics. For this alternate parameter set, excitatory conductance had rise time 0.25 ms, decay time 0.4 ms, peak conductance 1.6 nS, and release probability 0.82. Inhibitory conductance had rise time 0.35 ms, decay time 0.7 ms, peak conductance 3.9 nS, and release probability 0.92. Results obtained using these synaptic inputs were broadly similar to the result presented in *Figure 7*. Faster inhibitory conductances narrowed the width of ITD tuning curves, see *Figure 7—figure supplement 2*.

## Spike probability

To make direct comparisons to neural recordings in which we measured spike probability in response to clicks, we simulated responses of the model to synaptic inputs (described above) and computed spike probability semi-analytically. Spiking dynamics in the model are deterministic, thus the probability of spiking is determined completely by the probabilistic description of the synaptic inputs. Let the random variables $b_{N_E,p_E}$ and $b_{N_I,p_I}$ be independent, binomial-distributed random variables that represent the numbers of active excitatory and inhibitory inputs, respectively. Then the following defines the probability of a spike

$$\mathrm{P}(spike|ITD) = \sum_{n_I=0}^{\mathrm{N_I}} \sum_{n_E=0}^{\mathrm{N_E}} \mathbb{1}(n_E, n_I, ITD)\mathrm{P}\left(b_{N_E,p_E} = n_E\right)\mathrm{P}\left(b_{N_I,p_I} = n_I\right)$$

The indicator function $\mathbb{1}(n_E, n_I, ITD)$ takes values of 0 (no spike) or 1 (spike). These binary values were determined by simulating the model in response to all possible input combinations at the specified ITD value and recording whether the AIS-voltage crossed a fixed threshold (-20 mV) or not.

## Acknowledgements

We thank Anna Thiessen for her help in performing the electron microscopic analysis, Dr. Eric Verschooten for help with recording and analysis software. We also thank Dr. Kenneth Ledford for programming expertise in dynamic clamp experiments.

## Additional information

### Funding

| Funder | Grant reference number | Author |
|---|---|---|
| Fonds Wetenschappelijk Onderzoek | Ph.D. fellowship | Tom P Franken |
| Bijzonder Onderzoeksfonds | OT-14-118 | Philip X Joris |
| Fonds Wetenschappelijk Onderzoek | G.0961.11 | Philip X Joris |
| Fonds Wetenschappelijk Onderzoek | G.0A11.13 | Philip X Joris |
| Fonds Wetenschappelijk Onderzoek | G.091214N | Philip X Joris |
| National Institute on Deafness and Other Communication Disorders | DC006212 | Philip H Smith Philip X Joris |
| National Institute on Deafness and Other Communication Disorders | DC011403 | Nace L Golding Philip X Joris |
| National Institute on Deafness and Other Communication Disorders | DC006788 | Nace L Golding |
| National Institute on Deafness and Other Communication | 1F31DC017377-01 | David B Haimes |

| Disorders | | |
| --- | --- | --- |
| National Science Foundation | DMS - 1951436 | Joshua H Goldwyn |

The funders had no role in study design, data collection and interpretation, or the decision to submit the work for publication.

## Author contributions
Tom P Franken, Conceptualization, Software, Formal analysis, Funding acquisition, Investigation, Visualization, Methodology, Writing - original draft; Brian J Bondy, Software, Formal analysis, Investigation; David B Haimes, Software, Formal analysis, Funding acquisition, Investigation, Visualization; Joshua H Goldwyn, Software, Formal analysis, Investigation, Visualization, Methodology, Writing - review and editing; Nace L Golding, Resources, Supervision, Funding acquisition, Methodology, Writing - review and editing; Philip H Smith, Resources, Formal analysis, Supervision, Funding acquisition, Investigation, Visualization, Methodology, Writing - review and editing; Philip X Joris, Conceptualization, Resources, Supervision, Funding acquisition, Investigation, Methodology, Writing - original draft

## Author ORCIDs
Tom P Franken (ID) https://orcid.org/0000-0001-7160-5152
Joshua H Goldwyn (ID) https://orcid.org/0000-0001-5733-9089
Nace L Golding (ID) http://orcid.org/0000-0003-4072-310X
Philip H Smith (ID) https://orcid.org/0000-0003-3841-7031
Philip X Joris (ID) https://orcid.org/0000-0002-9759-5375

## Ethics
Animal experimentation: This study was performed in accordance with the recommendations in the Guide for the Care and Use of Laboratory Animals of the National Institutes of Health. All in vivo procedures were approved by the KU Leuven Ethics Committee for Animal Experiments (protocol numbers P155/2008, P123/2010, P167/2012, P123/2013, P005/2014). All in vitro recording and immunohistochemistry experiments were approved by the University of Texas at Austin Animal Care and Use Committee in compliance with the recommendations of the United States National Institutes of Health.

## Decision letter and Author response
Decision letter https://doi.org/10.7554/eLife.62183.sa1
Author response https://doi.org/10.7554/eLife.62183.sa2

# Additional files
## Supplementary files
• Source code 1. Code to generate figures of the computational model.

• Transparent reporting form

## Data availability
Source data files have been provided for all figures with electrophysiological recordings and for the computational model (Figures 1,2,3,4,7,8 and supplements). Custom MATLAB code for the computational model is provided as a source code file.

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
