## [Decision Letter]

**Acceptance summary:**

Coding of sound location is thought to occur in the medial (MSO) and lateral superior olivary (LSO) nuclei, with the former devoted to interaural temporal cues and the latter devoted to level difference cues. The current study shows this dichotomy is too simplistic. The authors made whole cell recordings in vivo from principal cells of the LSO and show sensitivity to transient time difference stimuli. They revealed the basis for that sensitivity, which is the convergence of excitatory synaptic events with well-timed synaptic inhibition. The reviewers were unanimously enthusiastic about the data on the responses of the LSO to transient stimuli, and the important differences between the principal and non-principal LSO cell populations. In a second section, the authors used brain slice recordings to mimic the in vivo work by stimulating excitatory and inhibitory inputs. When they could not reproduce the in vivo findings by injecting inhibitory conductances through their electrode into the soma, they suggested that the inhibition was not somatic but axonal. This was supported by a model and anatomical observations.

**Decision letter after peer review:**

Thank you for submitting your article "Glycinergic axonal inhibition subserves acute spatial sensitivity to sudden increases in sound intensity" for consideration by *eLife*. Your article has been reviewed by 3 peer reviewers, and the evaluation has been overseen by a Reviewing Editor and Andrew King as the Senior Editor. The following individuals involved in review of your submission have agreed to reveal their identity: Dan H Sanes (Reviewer #1); Steve Colburn (Reviewer #3).

The reviewers have discussed the reviews with one another and the Reviewing Editor has drafted this decision to help you prepare a revised submission.

Summary:

Coding of sound location is thought to occur in the medial (MSO) and lateral superior olivary (LSO) nuclei, with the former devoted to interaural temporal cues and the latter devoted to level difference cues. The current study shows this dichotomy is too simplistic. The authors made whole cell recordings in vivo from principal cells of the LSO and show sensitivity to transient time difference stimuli. They revealed the basis of that sensitivity, which is the convergence of excitatory synaptic events with well-timed synaptic inhibition. The reviewers were unanimously enthusiastic about the data on the responses of the LSO to transient stimuli, and the important differences in the principal and non-principal LSO cell populations. In a second section, the authors used brain slice recordings to mimic the in vivo work by stimulating excitatory and inhibitory inputs. When they could not reproduce the in vivo findings by injecting inhibitory conductances through their electrode into the soma, they speculated that the inhibition is not somatic but axonal. In anatomical work, they provided some evidence for such axonal inhibition.

Essential revisions:

The in vitro work was not as compelling as the in vivo work. One issue was that the assumptions that went into setting up the experimental paradigm were not as clearly stated and justified for the in vitro experiments. Further, the anatomical work was also not quantitative, so it was hard to know if the axonal synapses were in fact responsible for the physiological results. We propose that you either remove the in vitro and anatomical section or provide more detail for that work so that the point you are trying to make is less conjectural. We prefer the second option.

(1) The reviewers' major concern relates to your hypothesis that inhibitory synapses on the axon initial segment provide a super-fast elevation of the threshold at the spike generation region.

We like the hypothesis but are not completely convinced. We suggest that you clarify the in vitro dynamic clamp assumptions and caveats, and quantify the anatomical observations. We agree that new benchwork is undesirable, but you might have existing data that would help. We favor the idea of a model that can be accomplished remotely. There's a published two-compartment model (AIS + soma/dendrite) that you could co-opt (https://pubmed.ncbi.nlm.nih.gov/30830905/).

Please add more detail on your choices of stimuli and on other aspects of the methods. In our view of your results, when you could not reproduce the in vivo findings by injecting inhibitory conductances through the electrode into the soma, you hypothesized that inhibition is not somatic but axonal. Nevertheless, the in vitro work was not as compelling as the in vivo work. The contrast between dynamic clamp of IPSPs with synaptically evoked IPSPs and their relative ability to mediate 'ITD' sensitivity is absolutely key (in your rationale) to the claim that AIS inhibition is what endows the system with its impressive behavior. You found that injection of somatic conductances were unable to shunt the EPSPs sufficiently, whereas the synapse was able to do so quite well. That, combined with the observation that inhibitory boutons are found on the AIS, is the meat of your argument. But there are problems: First, the dynamic clamp essentially reports a negative result, failure to mimic the synapse. This forces the reader to wonder if you went about it right. The methods share little about how you chose the conductance values and time course. Nothing is cited and details are sparse. You wrote that you tuned it up to get a certain size of IPSP, but how do we know if that is the right metric? Perhaps you were comparing the IPSPs to those reported in vivo. But here is another problem: we are not sure if you can compare in vitro and in vivo recordings in a quantitative way. in vitro, we see a certain resting potential of around -60 mV and overshooting spikes. in vivo, the resting potentials are hidden, and the traces all baselined to zero. The spikes are small. We do not know how the input resistances compare. Thus, the size of the IPSP is not a reliable metric for matching in vivo and in vitro, as little else matches. Another mystery is how the strength of the synaptic events was set, and how they are compared to the dynamic clamp IPSPs. A better approach would have been to measure the effective synaptic IPSC in voltage clamp in order to determine the conductance and the waveform, then use that to set the dynamic clamp values. With respect to the dynamic clamp, would there be some relationship between the ability of dynamic clamp IPSGs to block spikes and the distance of the recording electrode to the AIS? Were reconstructions obtained for the relatively effective cases (Figure 4—figure supplement 1D, bottom right)?

We note that the discussion hardly mentions the AIS work, suggesting that you may also not be that comfortable with your hypothesis. Furthermore, the anatomical work too was not quantitative, so it is hard to know if the axonal synapses were in fact responsible for the physiological results. Another concern related to the AIS anatomy is that it is hard to tell why LSO neurons are so heavily encrusted with inhibitory synapses if the ones that matter are a few boutons on the AIS.

(2) In vivo stimuli: We appreciate the detail provided, but some statements do not appear to be clearly supported by the evidence. For example, measures of ITD function half-widths are said to be narrower for LSO principal cells (lines 129-133; Figure 1K). The p value is modest and the data suggest a more nuanced interpretation. The population of LSO principal neurons is dominated by those with high CFs, and the opposite is true for MSO neurons. Furthermore, non-principal neuron half-widths are just as sharp as those of principal neurons for CFs above 4 kHz. Please clarify if this effect is due to better afferent entrainment to the higher frequency components of the click stimulus, to principal neuron biological factors, or both? Did you measure an FFT of the click stimulus? In addition, might the relatively poor performance of low CF neurons be related to non-uniform glycine receptor density, with greater levels in the medial limb (https://pubmed.ncbi.nlm.nih.gov/2890726/)?

The recorded LSO principal cells have ITD functions that are positioned outside of the relevant 120 µs ITD range for gerbils (Figure 1-figure suppl 2). We find it difficult to visualize the behaviorally-relevant contribution of the new property. The final experiment (Figure 7L and Figure 7-figure suppl 2) appears to show that ILD performance alone (ITD=0) is better than ITD performance alone (ILD=0), and when both properties are co-varied, the neurons still respond only at values that correspond to a fully lateralized stimulus position. Is this correct?

(3) The monaural suprathreshold SPL values were obtained as described in the methods (lines 788-798), but it is not always clear which SPL values were used to obtain click-evoked ITD functions (Figure 1—figure supplement 1 provides levels for tone-evoked ILDs, and Figure 2 states that the SPL was 60 dB, presumably at both ear). Please provide information about the effect of monaural sound level on the click-evoked ITD functions, including whether this factor has an impact on the relevant range of time differences. Specifically, while ILD and ITD were varied independently and together (Figure 7), was ITD ever adjusted to be appropriate for each ILD presented (i.e., within the +/- 120 µs window)? If we understand correctly, the Voronoi diagram in Figure 7K suggests that the neuron does not fire in a biologically relevant range of ILDs (=/1 15 dB) and ITDs (+/- 120 µs) for a CF of 3.5 kHz.

(4) Discussion: Please offer us a more rigorous argument about the conditions under which LSO ITD processing is used, either alone, or alongside IID processing. Specifically, we remain uncertain about the contribution of the new LSO coding property to perception. For example, we know that humans have good single-click ILD thresholds (https://pubmed.ncbi.nlm.nih.gov/21117758/; https://pubmed.ncbi.nlm.nih.gov/15109699/), suggesting that these are sufficient to support rapid lateralization.

You refer briefly to sounds that would take advantage of the new LSO neuron encoding attribute (e.g., ln 17: "Locomotion generates adventitious sounds…"; lines 68-69: High-frequency transients are generated as adventitious sounds created by the locomotion of animals at close range…"). The authors should provide a referenced argument about the prevalence and acoustic properties of the natural sounds that could not be lateralized from their ITDs by MSO neurons, or from their ILDs by LSO neurons. Related to this is the final conclusion (lines 716-717, "…the time-based role of LSO is the more dominant (but previously underestimated) role."). Doesn't this statement rest on the premise that brief transients are the more dominant signal? The natural acoustic world is dominated by sounds with slow envelopes, especially for vocal communication. Consider reviews like Elliott and Theunissen (2009) and Ding et al. (2017). Gerbils might spend much of their time listening to, and possibly localizing, conspecific vocalizations which tend to be modulated in the same low range (Ter-Mikaelian et al., 2012), and which will be encoded by LSO neurons using a mechanism that is not exquisitely sensitive to microsecond time differences.

---

## [Author Response]

Essential revisions:The in vitro work was not as compelling as the in vivo work. One issue was that the assumptions that went into setting up the experimental paradigm were not as clearly stated and justified for the in vitro experiments. Further, the anatomical work was also not quantitative, so it was hard to know if the axonal synapses were in fact responsible for the physiological results. We propose that you either remove the in vitro and anatomical section or provide more detail for that work so that the point you are trying to make is less conjectural. We prefer the second option.(1) The reviewers' major concern relates to your hypothesis that inhibitory synapses on the axon initial segment provide a super-fast elevation of the threshold at the spike generation region.We like the hypothesis but are not completely convinced. We suggest that you clarify the in vitro dynamic clamp assumptions and caveats, and quantify the anatomical observations. We agree that new benchwork is undesirable, but you might have existing data that would help. We favor the idea of a model that can be accomplished remotely. There's a published two-compartment model (AIS + soma/dendrite) that you could co-opt (https://pubmed.ncbi.nlm.nih.gov/30830905/).

We have followed the suggestion of the reviewers, and have included a computational model in the revised version of the manuscript. This allowed us to verify whether inhibitory synapses added to the AIS have a more powerful effect than when they are added to the soma, as was suggested by the in vitro recordings. These data are presented in a new paragraph with a new figure (new Figure 7, with two figure supplements). The results show indeed that adding synapses to the AIS results in an outsized effect of pronounced spike suppression compared to the situation where these inhibitory synapses are added to the soma, and that this occurs for a wide range of parameters. We have also added further details about the dynamic clamp experiments in the Methods section and how they are interpreted in the Results. We have also added specific citations to the literature (also specified below) that provided us with the synaptic input parameters that form a critical part of these experiments. We reply below to the question regarding the anatomical observations.

Please add more detail on your choices of stimuli and on other aspects of the methods. In our view of your results, when you could not reproduce the in vivo findings by injecting inhibitory conductances through the electrode into the soma, you hypothesized that inhibition is not somatic but axonal. Nevertheless, the in vitro work was not as compelling as the in vivo work. The contrast between dynamic clamp of IPSPs with synaptically evoked IPSPs and their relative ability to mediate 'ITD' sensitivity is absolutely key (in your rationale) to the claim that AIS inhibition is what endows the system with its impressive behavior. You found that injection of somatic conductances were unable to shunt the EPSPs sufficiently, whereas the synapse was able to do so quite well. That, combined with the observation that inhibitory boutons are found on the AIS, is the meat of your argument. But there are problems: First, the dynamic clamp essentially reports a negative result, failure to mimic the synapse. This forces the reader to wonder if you went about it right. The methods share little about how you chose the conductance values and time course. Nothing is cited and details are sparse. You wrote that you tuned it up to get a certain size of IPSP, but how do we know if that is the right metric? Perhaps you were comparing the IPSPs to those reported in vivo. But here is another problem: we are not sure if you can compare in vitro and in vivo recordings in a quantitative way. in vitro, we see a certain resting potential of around -60 mV and overshooting spikes. in vivo, the resting potentials are hidden, and the traces all baselined to zero. The spikes are small. We do not know how the input resistances compare. Thus, the size of the IPSP is not a reliable metric for matching in vivo and in vitro, as little else matches. Another mystery is how the strength of the synaptic events was set, and how they are compared to the dynamic clamp IPSPs. A better approach would have been to measure the effective synaptic IPSC in voltage clamp in order to determine the conductance and the waveform, then use that to set the dynamic clamp values. With respect to the dynamic clamp, would there be some relationship between the ability of dynamic clamp IPSGs to block spikes and the distance of the recording electrode to the AIS? Were reconstructions obtained for the relatively effective cases (Figure 4—figure supplement 1D, bottom right)?

We chose values of IPSC/G dynamics based on a published voltage-clamp study of IPSC kinetics in gerbils (Walcher et al. 2011), and we now cite this study in the Methods when describing stimulus construction. We also include more details in the Methods regarding dynamic clamp performance, and note that this system has been fully described in publication (Yang et al., 2015), now cited. It should also be mentioned that data collection for dynamic clamp experiments is technically demanding: it takes time to find the right parametric conditions to properly simulate ITD tuning in vitro, and many repetitions are required at different time differences. The stability of the recordings is critical, and this is facilitated in recordings from animals of ages where IPSG kinetics are nearly, though not fully mature. Although the kinetics of the PSPs were set to match values at the ages we have used (~P20), we have used the computational model to examine the effects of faster, fully mature kinetics of inhibition on ITD tuning (new Figure 7—figure supplement 2). The results of these simulations show that, as expected, faster kinetics of IPSGs result in narrower tuning functions, but the differential (stronger) influence of axonal IPSPs. vs their somatic counterparts remains.

The amplitude of synaptically evoked IPSPs (shocks) was the same as those created through dynamic clamp: the strength of stimulation was adjusted to achieve 5-10 mV hyperpolarizations (similar as IPSPs evoked in vivo for an average resting membrane potential of -56.9 mV (Methods)). Given the similar resting potentials (~-60 mV) and identical pipette solutions in vitro, comparable driving forces for inhibition were achieved. Intrinsic membrane properties (V_rest_, membrane time constant, input resistance) have reached an asymptote by P18 in gerbils (Walcher et al., 2011), further supporting the validity of comparing our in vitro and in vivo data sets. Unfortunately, we cannot selectively control stimulation of only axonal synapses. However, our explanation that somatic inhibition is subject to additional attenuation compared to inhibition from synapses in the hillock or in the axon itself is biophysically plausible and is now additionally supported by neuron modeling.

We do not expect there to be a relationship between recording site and the location of the AIS. The AIS emerges from the soma in LSO neurons, and given the input resistance of LSO cells (~40 MΩ), we would expect there to be little attenuation across the two ends of the soma. We did not obtain anatomical labeling in dynamic clamp because the long recording times reduce the probability of obtaining clean pull-offs of the recording pipette, which usually degrades the quality of the anatomy.

We note that the discussion hardly mentions the AIS work, suggesting that you may also not be that comfortable with your hypothesis. Furthermore, the anatomical work too was not quantitative, so it is hard to know if the axonal synapses were in fact responsible for the physiological results. Another concern related to the AIS anatomy is that it is hard to tell why LSO neurons are so heavily encrusted with inhibitory synapses if the ones that matter are a few boutons on the AIS.

We would like to clarify that we are not suggesting that the somatic inhibition is not important. Instead, the results from the computational model indicate that, in the presence of somatic inhibition, even a few boutons on the AIS have an outsized effect (new Figure 7A, solid line versus dashed line). In addition, a similar number of synapses on the AIS without somatic inhibition are not sufficient (new Figure 7A, solid line versus dotted line). Both somatic as well as AIS synapses are thus important. We included a sentence in our description of modeling results to make this point clearer.

We are comfortable with our hypothesis and the AIS data in our manuscript. The reason that the Discussion of the initial manuscript did not dwell on the AIS work is that we found it particularly important to elaborate on the wider implications of our data towards a new view of the LSO, and of course there are length constraints. We have now elaborated on the AIS paragraph in the Discussion to emphasize its broader significance.

We are not able to quantify the light or electron microscopic data. Serial EM reconstructions across such a broad area would be prohibitively time consuming and beyond the scope of the study. For the triple immunolabeling data, we cannot provide a large-scale, unbiased quantification because optimal imaging conditions were only achieved for a small subset of axons that ran parallel to the surface of the section, and only a subset of this smaller population could be verified to be completely intact beyond the end of the AIS. We have attempted to provide as many of these clear examples as possible to convey (albeit qualitatively) that axonal inhibitory contacts are not uncommon in LSO neurons.

(2) In vivo stimuli: We appreciate the detail provided, but some statements do not appear to be clearly supported by the evidence. For example, measures of ITD function half-widths are said to be narrower for LSO principal cells (lines 129-133; Figure 1K). The p value is modest and the data suggest a more nuanced interpretation. The population of LSO principal neurons is dominated by those with high CFs, and the opposite is true for MSO neurons. Furthermore, non-principal neuron half-widths are just as sharp as those of principal neurons for CFs above 4 kHz. Please clarify if this effect is due to better afferent entrainment to the higher frequency components of the click stimulus, to principal neuron biological factors, or both? Did you measure an FFT of the click stimulus? In addition, might the relatively poor performance of low CF neurons be related to non-uniform glycine receptor density, with greater levels in the medial limb (https://pubmed.ncbi.nlm.nih.gov/2890726/)?

Looking into this, we discovered that due to the choice of X axis limits, two MSO data points with high CF – which were included in the analysis – were not visible in Figure 1K. We apologize for this oversight and we have corrected this in the revised version of the manuscript. These are not shown in Figure 1L because they did not have the required number of repetitions (ITD-SNR was only calculated for data sets with at least 10 repetitions, as specified in Methods).

We indeed find a strong correlation between halfwidth of ITD-tuning and CF. This is what one expects from a general systems point of view (increasing bandwidth of frequency tuning with increasing CF), and could also be related to the point made by the Reviewers re. the non-uniform distribution of glycine receptors. We have added this reference to the manuscript.

We agree with the Reviewers that the differences in halfwidth between groups are not very strong. The functions in panels 1A,1C,1E and 1I show that halfwidth does not fully capture the difference between click ITD functions from LSO cells and MSO cells: the MSO tuning functions are much poorer shaped and noisier. Halfwidth of MSO functions may seem ‘good’ but the noisiness of the functions suggests that this metric does not fully capture the quality of tuning. In contrast, the ITD-SNR, which includes the variability of the responses, is a more complete quantification of ITD tuning, and we find a much more pronounced difference between principal LSO and MSO tuning functions, across the range of frequency tuning. Thus halfwidth underestimates the difference in quality of tuning between principal LSO cells and MSO cells. The reason we still kept these results is that it is an easier (and oft-used) metric than ITD-SNR. We added text in this section of Results to explain this better.

The ITD functions of non-principal LSO cells are more variable than the principal LSO cells. This is not surprising, as it is also morphologically and physiologically a heterogeneous group (Franken et al., 2018). Importantly, even at high CF, ITD-SNR is lower for non-principal than for principal neurons. Similar to the MSO functions, the functions for non-principal LSO neurons are noisier. In Author response image 1 we show unsmoothed click-ITD functions for the subgroup of principal and non-principal LSO neurons that are tuned above 4 kHz. Again, ITD-SNR captures the quality of tuning better than halfwidth. Additional analysis showed that ITD-SNR was significantly different for principal LSO cells and non-principal LSO cells. We have added this analysis to the manuscript.

**Author response image 1. sa2fig1:** 

The click stimulus was a rectangular rarefaction pulse of 20 μs width, scaled in amplitude according to the calibration for each animal. We clarified this in Methods. The spectrum is thus known, as well as the frequency calibration at the ear.

The recorded LSO principal cells have ITD functions that are positioned outside of the relevant 120 µs ITD range for gerbils (Figure 1-figure suppl 2). We find it difficult to visualize the behaviorally-relevant contribution of the new property. The final experiment (Figure 7L and Figure 7-figure suppl 2) appears to show that ILD performance alone (ITD=0) is better than ITD performance alone (ILD=0), and when both properties are co-varied, the neurons still respond only at values that correspond to a fully lateralized stimulus position. Is this correct?

Our response to these remarks is at several levels.

First, at the data level, responses for combined ITD and IID variations are available for only a few cells, as this requires very long recordings. One cannot simply use extracellular recordings since, as discussed in our previous work (Franken et al., 2018), these methods undersample principal LSO cells and click stimuli make spike isolation difficult because of large mass potentials. We think that the most important finding in Figure 8/Figure 8—figure supplement 2 (note that the numbering of this figure has changed in the revised manuscript after inserting the figure on the computational model) is that the estimated functions where both click ITD and click ILD vary in a physiological way (magenta) results in steeper tuning functions of which the slope can be positioned in the physiological range (Figure 8 suppl 2B). We have added another example cell from a juxtacellular recording, showing the same phenomenon (Figure 8 suppl 2C).

Second, it is very difficult to make assertions about the “relevant ITD range” (cf. Joris and van der Heijden 2019). Ultimately, LSO recordings to adventitious sounds in a natural environment are needed. Our stimuli are a very impoverished approximation of such sounds. Not only are our stimuli simple rectangular clicks delivered straight to the ear canal, but also both bullas are opened to allow access to the brainstem for the whole-cell recordings. Any effects of pinnae, head, torso, ground plane, etc. are absent. And of course the animal is under anesthesia, likely with an ineffective efferent system. The reviewers may be correct that this system may only come into play at fully lateralized angles, but with the physiological and acoustical data available, we just cannot know. We added some sentences to this effect to the Discussion.

There is a third, more subtle level bearing on this issue. “The new property” that the reviewers refer to is, in brief, the temporal acuity of LSO principal neurons to transient stimuli. That property is present, independent of which cues are varied. Even if only the IID cue is varied, the various temporal specializations come into play. The ideal experiment would be one where temporal specializations (large axosomatic synapses in afferent pathways, differential conduction speeds, leaky membranes, AIS inhibition,…) could be turned on and off while measuring sensitivity to the different binaural cues in response to different types of sounds. Our prediction is that binaural sensitivity to transients would be degraded by such manipulations, but of course such manipulations are experimentally out of reach. An interesting experiment, which is likely possible not too far into the future, would be to turn off the different LSO populations while measuring behavioral sensitivity to transient and sustained stimuli (an older study in this spirit is by Li and Kelly (1992), where they showed remaining IID-sensitivity in the inferior colliculus to tones after kainic acid lesions of the SOC). In summary, studying responses to ITDs and IIDs should not be equated to studying the system in a state where temporal factors come into play vs. one where they don’t come into play (see also our reply to (4) below).

(3) The monaural suprathreshold SPL values were obtained as described in the methods (lines 788-798), but it is not always clear which SPL values were used to obtain click-evoked ITD functions (Figure 1—figure supplement 1 provides levels for tone-evoked ILDs, and Figure 2 states that the SPL was 60 dB, presumably at both ear). Please provide information about the effect of monaural sound level on the click-evoked ITD functions, including whether this factor has an impact on the relevant range of time differences. Specifically, while ILD and ITD were varied independently and together (Figure 7), was ITD ever adjusted to be appropriate for each ILD presented (i.e., within the +/- 120 µs window)? If we understand correctly, the Voronoi diagram in Figure 7K suggests that the neuron does not fire in a biologically relevant range of ILDs (=/1 15 dB) and ITDs (+/- 120 µs) for a CF of 3.5 kHz.

Indeed the sound level in Figure 2D was 60 dB SPL at both ears, we specified this in the figure legend.

We provide a new figure that shows the effect of monaural sound level (Figure 1—figure supplement 3). The data shows that differences in sound levels do not explain the differences in ITD-SNR between cell types (Figure 1- figure suppl. 3C,D). We also followed the Reviewers’ other suggestion and show the effect of sound level on the shape of ITD functions of principal LSO neurons (Figure 1- figure suppl. 3A,B). The data shows that higher levels can result in steeper slopes and narrower halfwidths, bringing the physiological (left) side of the slope closer to 0. Regarding appropriate combinations of ITD and ILD and physiological range, we refer to our reply above on the role of ITD, ILD and combination of both. In the experiment in Figure 8, both cues are varied independently. Thus the appropriate pairs are a subset of these combinations, which are estimated for each unit as the magenta functions in Figure 8 and Figure 8—figure supplement 2.

(4) Discussion: Please offer us a more rigorous argument about the conditions under which LSO ITD processing is used, either alone, or alongside IID processing. Specifically, we remain uncertain about the contribution of the new LSO coding property to perception. For example, we know that humans have good single-click ILD thresholds (https://pubmed.ncbi.nlm.nih.gov/21117758/; https://pubmed.ncbi.nlm.nih.gov/15109699/), suggesting that these are sufficient to support rapid lateralization.

We refer to our remark above that temporal specializations in the LSO benefit sensitivity to transients both for ITDs and IIDs, but we acknowledge that we have to present our argument more clearly. We added the following text in the Discussion. “Our assertion is not that there is no IID-sensitivity in LSO, which has been abundantly demonstrated both to sustained and transient sounds, or that ITD is the only important binaural cue. […] Our recordings show directly that IIDs affect both amplitude and timing of EPSPs and IPSPs (Figure 8).”

You refer briefly to sounds that would take advantage of the new LSO neuron encoding attribute (e.g., ln 17: "Locomotion generates adventitious sounds…"; lines 68-69: High-frequency transients are generated as adventitious sounds created by the locomotion of animals at close range…"). The authors should provide a referenced argument about the prevalence and acoustic properties of the natural sounds that could not be lateralized from their ITDs by MSO neurons, or from their ILDs by LSO neurons. Related to this is the final conclusion (lines 716-717, "…the time-based role of LSO is the more dominant (but previously underestimated) role."). Doesn't this statement rest on the premise that brief transients are the more dominant signal? The natural acoustic world is dominated by sounds with slow envelopes, especially for vocal communication. Consider reviews like Elliott and Theunissen (2009) and Ding et al. (2017). Gerbils might spend much of their time listening to, and possibly localizing, conspecific vocalizations which tend to be modulated in the same low range (Ter-Mikaelian et al., 2012), and which will be encoded by LSO neurons using a mechanism that is not exquisitely sensitive to microsecond time differences.

Tying LSO function to the lateralization of adventitious sounds is a multilayered argument, which is extensively discussed by Joris and Trussell (2018). Although it would be easy to add a paragraph summarizing these arguments, we feel it would take too much space and would be redundant. We limited ourselves to adding several sentences and references in the Introduction.

We used ‘dominant’ in the conclusion because principal cells are the most common cell type in the LSO. The lack of acoustic field recordings prevents making a statement re. the relative importance of slow envelope modulation *versus* transients for small critters like gerbils. In general, the physiological literature is strongly biased towards communication sounds when discussing functional relevance, and shows an almost complete absence of consideration of adventitious sounds. We share the view of the reviewers that slow-envelope cues are particularly important for the coding of communication sounds. However, the existing literature (referenced in the manuscript) shows that such envelopes are not very potent binaural cues in the LSO. As argued in Joris and Trussell (2018), we find it hard to see how spatial aspects of vocal communication would have driven the extreme adaptations in the LSO-circuit.

We added a sentence in this paragraph to clarify for which sounds the time-based role is most important.